# Crystalline hydrogen bonding of water molecules confined in a metal-organic framework

Jinhee Bae [1], Sun Ho Park [1], Dohyun Moon [2✉] & Nak Cheon Jeong [1,3✉]

Hydrogen bonding (H-bonding) of water molecules confined in nanopores is of particular interest because it is expected to exhibit chemical features different from bulk water molecules due to their interaction with the wall lining the pores. Herein, we show a crystalline behavior of H-bonded water molecules residing in the nanocages of a paddlewheel metal-organic framework, providing in situ and ex situ synchrotron single-crystal X-ray diffraction and Raman spectroscopy studies. The crystalline H-bond is demonstrated by proving the vibrational chain connectivity arising between hydrogen bond and paddlewheel Cu−Cu bond in sequentially connected Cu–Cu·····coordinating $H_2O$·····H-bonded $H_2O$ and by proving the spatial ordering of H-bonded water molecules at room temperature, where they are anticipated to be disordered. Additionally, we show a substantial distortion of the paddlewheel $Cu^{2+}$-centers that arises with water coordination simultaneously. Also, we suggest the dynamic coordination bond character of the H-bond of the confined water, by which an H-bond transitions to a coordination-bond at the $Cu^{2+}$-center instantaneously after dissociating a previously coordinated $H_2O$.

[1] Department of Physics and Chemistry, DGIST, Daegu 42988, Korea. [2] Beamline Department, Pohang Accelerator Laboratory, Pohang 37673, Korea. [3] Center for Basic Science, DGIST, Daegu 42988, Korea. ✉email: dmoon@postech.ac.kr; nc@dgist.ac.kr

Hydrogen bond (H-bond) is one of the most intensively studied subjects in contemporary chemistry due to its high academic applicability in interpreting and predicting various critical phenomena observed in different fields, including chemistry[1–5], biology[6,7], physics[8–10], energy[11–14] and environmental science[15,16].

Nevertheless, revealing the H-bond dynamics of confined water molecules within the nanopores remains challenging because the confined water exhibits substantially altered chemical behaviour from bulk water. So far, zeolites and carbon nanotubes have been mainly examined to elucidate the H-bond of the confined water molecules[17–23], while the H-bond dynamics can depend upon the variety of chemical environments.

Metal-organic frameworks (MOFs) hold unique potential as structurally well-designed porous materials because they can translate their permanent porosity, modular topology and wide chemical variety into the tunability of their physicochemical functions. The tunability has recently accelerated the investigation of MOFs for potential use, particularly in proton conduction[24], adsorption-based heat pumping[25–29] and atmospheric water harvesting[30–36], where H-bond of confined water molecules can play a crucial role.

Dynamics of H-bond in MOFs, anticipated to be more sophisticated owing to the variability of metal ions, organic linkers and topologies, have rarely been studied[37–39]. Specifically, the H-bond network in MOF pores can be altered depending on the size and shape of pores, the components of wall lining pores and the presence of open metal sites (OMSs) residing at metal centres where water molecules can coordinate. Among these factors, the OMS can substantially influence the mobility of pore-filling water molecules due to their possible H-bonds with coordinating water molecules at the OMS. For instance, the water molecules H-bonded with the coordinated water can be more polarised than the unbound water molecules residing at the centre of pores. Therefore, the pore-filling water molecules around the OMSs can have a specific H-bond network along the interior walls. Recently, Paesani, Dincă and coworkers demonstrated the templating effect of H-bond, using a MOF containing cylindrical pores with cobalt OMS[37]. They observed that water molecules bind to the cobalt OMSs preferentially at low humidity and subsequently form disconnected one-dimensional H-bond chains bridging the cobalt ions at high humidity. Meanwhile, Walton and coworkers visualised the structural change of a Zn(II)-based paddlewheel MOF, which depends on the extent of water loading[38]. Based on difference envelope density calculation, they simulated preferential water sites in the pores: unbound water molecules prefer the sites near the hydrophilic framework at low water loading, but they transition to the centre of pores at high loadings. Since metal centres and metal-coordinating water molecules give rise to the hydrophilic environment and the templating effect, strong H-bonds must be in play between the metal-coordinating and pore-filling water molecules.

HKUST-1, a paddlewheel MOF also known as $Cu_3(BTC)_2$, where $BTC^{3-}$ is 1,3,5-benzenetricarboxylate (Fig. 1), have shown a unique structural feature with two types of large cages and a type of small cage. The small cages are packed with the fashion of primitive cube, connected by paddlewheel Cu–Cu nodes, and the OMSs at the Cu nodes are faced toward the large cages, where H-bonds between pore-filling and Cu-coordinating water molecules can be formed (See detailed topological description in the hypothesis section below). HKUST-1 have also shown a unique Cu–Cu stretching vibrational feature on Raman spectroscopy, depending on the $Cu^{2+}$-coordinating molecular species, such as water, methanol (MeOH) and ethanol (EtOH, Supplementary Section 1 and Supplementary Fig. 1)[40–44]. Considering that the Cu–Cu vibration is sensitive to the circumstance around the

$Cu^{2+}$ centre, the pore-filling water molecules can also be expected to influence the Cu–Cu vibration by their H-bonds, which can be thought of as a chain-connected second coordination sphere (Fig. 1e).

This work presents the crystalline behaviour of H-bonded (pore-filling) water molecules, demonstrating their vibrational chain connectivity to the paddlewheel Cu–Cu and ice-like spatial ordering in the nanocages, which must appear in a crystalline lattice generally. The chain connectivity was found from Raman studies to appear between sequentially connected paddlewheel Cu–Cu, coordinated $H_2O$ at the $Cu^{2+}$ center, and H-bonded $H_2O$ at the coordinated $H_2O$ (Cu–Cu···$H_2O$(coordinating)···$H_2O$(H-bonded); see Fig. 1e-ii). Thus, we show how the H-bond of pore-filling water molecule affects the Cu–Cu vibrational band. Considering that vibrational chain connectivity may arise in sequentially connected covalent bonds, the vibrational chain connectivity of H-bonded water molecules underscores that their H-bonds can behave like the chemical bond of the crystalline lattice, at least, in a specific environment such as MOF nanopores. In light of the high degree of freedom of H-bonded bulk water molecules in terms of their molecular motions at room temperature, one might anticipate that H-bonded water molecules in nanopores will also be highly disordered. However, in situ synchrotron single-crystal X-ray diffraction (SCXRD) results measured at 298 K provide compelling evidence that unbound H-bonded water molecules in nanopores prefer the sites near the coordinating water molecules with high ordering even at room temperature. The ice-like spatial ordering of the H-bonded water molecules, combined with their vibrational chain connectivity, thus, demonstrates the crystalline behavior of the H-bonded water molecules. In addition, we also present a distortion of the paddlewheel $Cu^{2+}$ center that appears after water coordination. Further, we suggest the dynamic coordination bond character of the confined H-bond, by which the H-bonded water molecule forms the coordination bond at $Cu^{2+}$ center instantaneously after dissociating the previously coordinated water molecule[45–47].

## Results and discussion

**Hypothesis.** HKUST-1 consists of two types of large cages with the shape of a truncated cube (type-I and -II cages) and a type of small cage with the shape of a truncated octahedron (type-III cage, Supplementary Section 2 and Supplementary Figs. 2-3). In a topological view, the link of type-III cages by the paddlewheel Cu centres in the fashion of a primitive cube constructs HKUST-1. Based on this topology, HKUST-1 has a unique structural feature: OMSs at the paddlewheel nodes face toward the open space of the type-I cages, which are large enough to accommodate multiply H-bonded water molecules.

With this structural insight in mind, we hypothesised the vibrational chain connectivity of the H-bond, conceiving that H-bond between the coordinating and pore-filling $H_2O$ molecules could sensitively influence the Cu–Cu vibrational mode. When coordinating $H_2O$ is absent initially, open-state paddlewheel $Cu^{2+}$ ions can be strongly associated together, exhibiting conceivably short bond length with high vibrational frequency (see Fig. 1c). Once $H_2O$ molecules coordinate at the $Cu^{2+}$ centres, however, the paddlewheel $Cu^{2+}$ ions can be loosely bound to each other, exhibiting conceivably longer bond length with lower vibrational frequency than that in the open state owing to the partial electron donation from coordinating $H_2O$ to an antibonding molecular orbital of $Cu^{2+}$ ion (Fig. 1d)[48–50]. (We note that, while the Raman spectra of open-state and $H_2O$-coordinating HKUST-1s have been reported previously[40–44], no report has been made on crystallographic Cu–Cu lengths varying with the $H_2O$-coordinations). We hypothesised further that when the

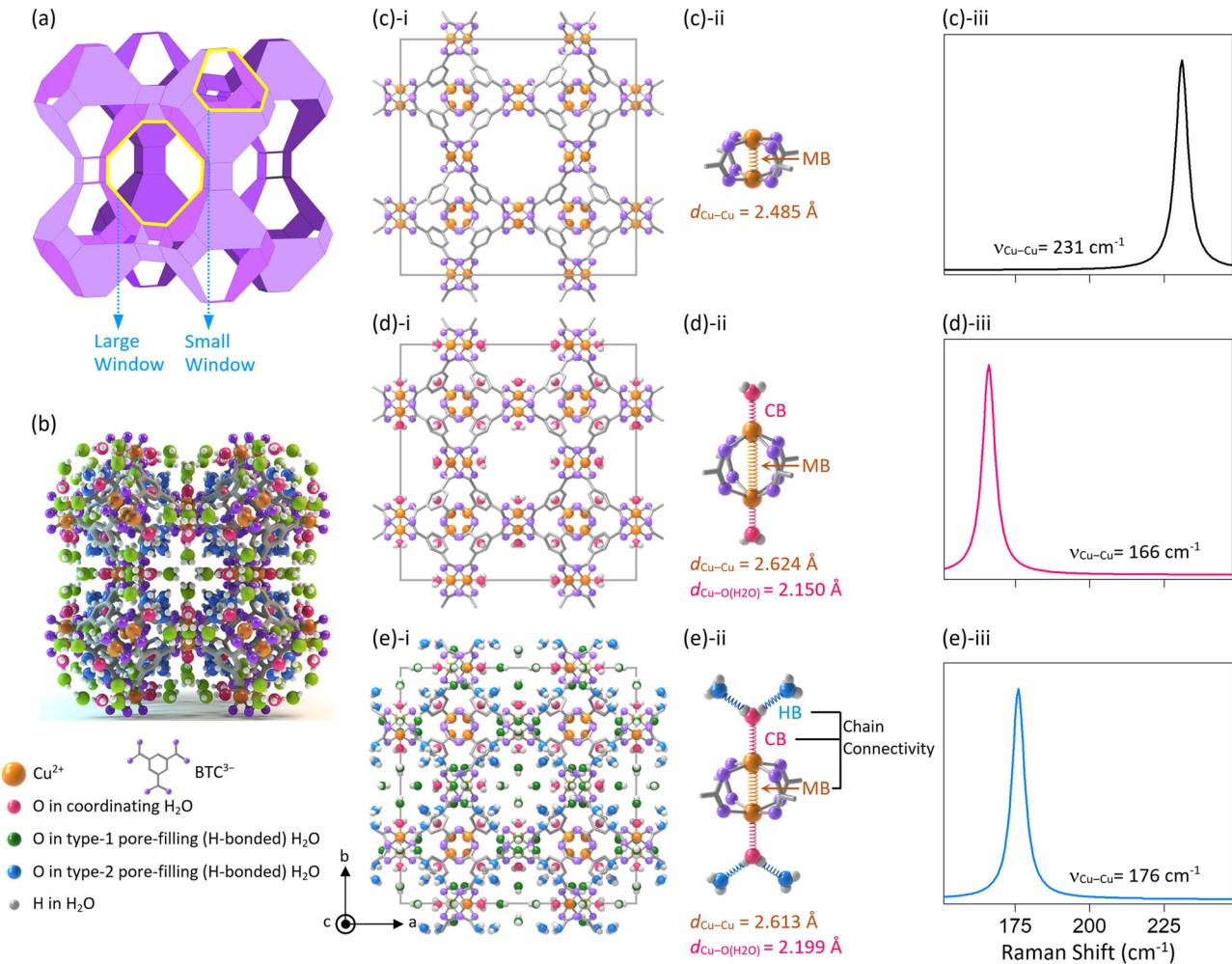

**Fig. 1 Hypothesis for the chain connectivity of hydrogen-bonded water molecules. a** Three-dimensional (3-D) schematic illustration of HKUST-1 structure. **b** 3-D view of the single-crystal structure of HKUST-1. **c–e** (**i**) Hypothetic crystal structures viewed along the *c* direction, (**ii**) illustrations of Cu–Cu paddlewheel node and (**iii**) estimated Raman signals of Cu–Cu stretching vibration, as for the HKUST-1 crystal under (**c**) activated, (**d**) only $H_2O$-coordinated and (**e**) $H_2O$-filled states.

coordinating $H_2O$ molecules begin to form H-bonds with pore-filling $H_2O$ molecules, the paddlewheel $Cu^{2+}$ ions can undergo strong association reversibly, exhibiting shortened bond length with higher vibrational frequency due to the weakened coordination strength by the chain connectivity of H-bonded water molecules (see Fig. 1e). Envisioning a strong association of H-bonded $H_2O$ to the coordinating $H_2O$, we further hypothesised that the H-bonded $H_2O$ molecules could site-selectively occupy the pores even at room temperature, differently from bulk $H_2O$ molecules that must be dynamic in their translational and rotational motions. On the other hand, the above hypothesis is difficult to demonstrate if a bulk experimental system such as a molecular $Cu_2(CH_3CO_2)_4$ complex dissolved in water is employed because, in an aqueous system, $H_2O$-coordination-free complex and only $H_2O$-coordinating complex without H-bonded water cannot be isolated. Thus, a water-endurable MOF, such as HKUST-1, with paddlewheel structure and permanent porosity is suitable for demonstrating the above hypothesis.

**Sample preparation**. We prepared activated HKUST-1 crystals that contain no water molecule (denoted as Act-HKUST-1). We also prepared $H_2O$-coordinating HKUST-1 crystals that are expected to contain coordinating $H_2O$ only (denoted as $H_2O^{[C]}$-HKUST-1, where [C] stands for coordinating) and $H_2O$-filled HKUST-1 crystals that contain both coordinating and pore-filling

$H_2O$ (denoted as $H_2O^{[F]}$-HKUST-1, where [F] stands for filling pores, see Methods). Further, we prepared MeOH- and EtOH-filling HKUST-1 crystals (denoted as $MeOH^{[F]}$-HKUST-1, and $EtOH^{[F]}$-HKUST-1, respectively) via soaking Act-HKUST-1 crystals into pure MeOH and EtOH, respectively. The phase purities of the samples were checked by examining [1]H nuclear magnetic resonance (NMR) spectroscopy and powder X-ray diffraction (PXRD, Supplementary Section 3 and Supplementary Fig. 4).

**Raman and crystallography studies at 220 K**. We conducted Raman studies with Act-HKUST-1, $H_2O^{[C]}$-HKUST-1 and $H_2O^{[F]}$-HKUST-1 initially at 220 K. While an Act-HKUST-1 exhibited a Raman band for Cu–Cu stretching vibration at 231 cm$^{-1}$, $H_2O^{[C]}$-HKUST-1 and $H_2O^{[F]}$-HKUST-1 exhibited the bands at 166 and 176 cm$^{-1}$, respectively, (Supplementary Section 4 and Supplementary Fig. 5). To investigate the correlation between the Cu–Cu vibrational band energies and their bond lengths upon the HKUST-1 states, we conducted synchrotron SCXRD experiments with the crystals at 220 K. As expected, the SCXRD results confirm that the Act-HKUST-1 crystal contains no water molecule (see Supplementary Section 5, Supplementary Figs. 6–8 and Supplementary Table 1; also see original crystal structures in Supplementary Data 1). On the other hand, we

expected that the $H_2O^{[C]}$-HKUST-1 would contain coordinating $H_2O$ only because the Cu–Cu Raman band appeared at the lowest frequency (166 cm$^{-1}$, see Fig. 1 and the hypothesis above). However, the SCXRD result reveals that the $H_2O^{[C]}$-HKUST-1 contains not only coordinating $H_2O$ (denoted as $H_2O^{[C]}$) but also pore-filling $H_2O$ with 40 mol% at specific positions in type-I cages (denoted as type-1 pore-filling $H_2O$ with the abbreviation of $H_2O^{[F]1}$; see Fig. 2a–c and Supplementary Table 2-3). This result indicates that unbound pore-filling $H_2O$ quite prefers the site near the coordinating $H_2O$. The appearance of the Raman band at the lowest frequency despite the inclusion of the $H_2O^{[F]1}$ is ascribed to the dynamic coordination bond character of H-bond, by which $H_2O^{[F]1}$ can associate at the $Cu^{2+}$ centre instantaneously after replacing the previous coordination bond of $H_2O^{[C]}$ on the molecular time scale[45–47]. SCXRD result of $H_2O^{[F]}$-HKUST-1, where pore-filling $H_2O$ molecules are supposed to bind at coordinating $H_2O$ via H-bonds strongly, shows a type of coordinating $H_2O$ and two types of pore-filling $H_2O$ (type-1 and type-2; denoted with the abbreviation of $H_2O^{[F]2}$, see Fig. 2d–f). This result describes accurate positions of the $H_2O$ molecules: coordinating $H_2O^{[C]}$ at the $Cu$–$O^{[C]}$ distance of 2.199 Å; pore-filling $H_2O^{[F]1}$ and $H_2O^{[F]2}$ at the $O^{[C]}$–$O^{[F]}$ distance of 2.825 and 3.065 Å, respectively, (Supplementary Table 3)[1–3]. Also, these crystallographic data support our hypothesis, providing the dependence of Cu–Cu lengths (2.485, 2.624 and 2.613 Å) upon the $Cu^{2+}$ states.

**Raman studies at various temperatures**. To observe the influence of temperature on vibrational energy, we also conducted the Raman experiments at 298 K, slowly removing $H_2O$ from an $H_2O^{[F]}$-HKUST-1 under vacuum conditions. As a result, the pattern of spectral changes at 298 K was the same as that observed at 220 K. Whereas the Raman band of $H_2O^{[F]}$-HKUST-1 appeared at 176 cm$^{-1}$, the band redshifted approaching 163 cm$^{-1}$ as the filled $H_2Os$ were removed from the pores (Supplementary Section 6 and Supplementary Fig. 9). However, the band was blueshifted to 231 cm$^{-1}$ in reverse after all coordinated $H_2Os$ were thoroughly removed from the sample. To ascertain the absence of temperature effect on the vibration, we further performed the Raman analysis at 170 and 250 K with an $H_2O^{[F]}$-HKUST-1 (Supplementary Section 7 and Supplementary Fig. 10). As predicted, the Cu–Cu Raman bands at both 170 and 250 K were the same as those measured at 220 and 298 K. However, as the temperature decreases, the O–H vibration of pore-filling $H_2O$ (3110–3146 cm$^{-1}$) redshifted simultaneously with band sharpening due to the ice H-bond. On the other hand, we examined an isotope effect on the vibrational band after introducing $D_2O$ and $H_2^{18}O$ into HKUST-1 crystals, respectively. However, no isotope effect was observed (Supplementary Section 8 and Supplementary Fig. 11). Thus, these Raman results explicitly support the absence of temperature and isotope effects on the chain connectivity.

**Raman studies with EtOH and MeOH**. We found that the chain connectivity is generic for EtOH and MeOH. EtOH$^{[F]}$-HKUST-1 and MeOH$^{[F]}$-HKUST-1 showed the Cu–Cu vibrational bands at 188 and 180 cm$^{-1}$, respectively, (Supplementary Section 9 and Supplementary Figs. 12-13). However, these bands redshifted to 178 and 172 cm$^{-1}$ when the pore-filling EtOH and MeOH were removed, respectively. After thorough removal of even coordinating EtOH and MeOH, both bands were rather blueshifted to 231 cm$^{-1}$. Meanwhile, EtOH and MeOH confined in nanopores have shown a behaviour different from their bulk states in terms of C–O and C–C vibrations (Supplementary Section 9-10 and Supplementary Figs. 12-14)[51]. Given that the binding of a

molecule governs its internal vibrational energies, the above vibrational changes can be ascribed to the enhanced H-bond strength of confined EtOH and MeOH molecules.

**In situ Raman at room temperature**. We wondered if the water-ingress will give a result similar to the above water-egress in Raman spectral changes. We performed in situ Raman experiments to address this question, exposing an Act-HKUST-1 crystal to moist air at room temperature. As a result, the in situ Raman spectra have shown a pattern similar to those observed in the above ex situ (water-egress) experiments (Fig. 3a and Supplementary Section 11 and Supplementary Figs. 15-16). The spectra show that the vibrational band of the Act-HKUST-1 redshifts from 231 to 163 cm$^{-1}$ as the exposure time increases. Whereas the band intensity at 231 cm$^{-1}$ gradually decreases, the band intensity at 163 cm$^{-1}$ gradually increases. A notable feature is that these bands are ratiometric at discrete positions rather than continuously shifting. This result indicates that although paddlewheel $Cu^{2+}$ centres are interconnected through $BTC^{3-}$ linkers, the Cu–Cu vibrations are not interconnected. When the exposure time increased more, by contrast, the band was blueshifted, approaching 176 cm$^{-1}$ in a continuous fashion rather than a discrete one. Also, the band was broadened as the quantity of filling $H_2O$ increased. We speculated that the continuous shift and broadening of the Raman band could be attributed to the increase in the number of H-bonds around a coordinating $H_2O$ (also see IR spectra in Supplementary Section 12 and Supplementary Fig. 17)[52]. Based on this speculation, we questioned how many H-bonds would be present around a coordinating $H_2O$ because the pattern of the band shift would be discrete if only one or two H-bonds are present.

To this end, we monitored the weight changes of Act-HKUST-1 crystals with a microbalance, exposing the crystals to moist air. The test showed a continuous increase in weight by 52.7 wt% (Fig. 3b). This weight change indicates that the stoichiometric molar ratio of $[H_2O]/[Cu^{2+}]$ is 5.9. We also noted an inflexion point in the curve that appeared at ~8.3 wt% because this value is very close to the value theoretically calculated for the $[H_2O^{[C]}]/[Cu^{2+}]$ ratio (1.0). Taking into account the above results, we concluded that the molar ratio of $[H_2O^{[F]}]/[H_2O^{[C]}]$ (the average number of H-bonds around $H_2O^{[C]}$) should be ~4.9. We also found that this ratio is close to a value (4.66$_6$) theoretically calculated based on in situ SCXRD results (vide infra; Supplementary Section 13 and Supplementary Table 4). Therefore, the seemingly continuous Raman band shift can be ascribed to a result that ~4.9 H-bonds are reflected.

Additionally, we monitored the colour changes of the crystals using an optical microscope. While the colour of the Act-HKUST-1 crystal was a deep navy blue, the colour was turned to pale blue as the water exposure time increases (Fig. 3c and Supplementary Section 11 and Supplementary Fig. 16). Using UV-vis absorption spectroscopy, we confirmed that the colour change comes from the changes in the d-d transition band of $Cu^{2+}$ centres (see Fig. 3d and Supplementary Section 14 and Supplementary Fig. 18)[40–42].

**In situ SCXRD at 298 K**. We conducted in situ SCXRD experiments, exposing an Act-HKUST-1 crystal to moist air at 298 K to observe the correlation between the vibrational chain connectivity and Cu–Cu length. SCXRD results show that while the Cu–Cu length of the Act-HKUST-1 was 2.484 Å, the length was gradually increased as the exposure time increased and subsequently maximised to 2.632 Å when the contents of $H_2O^{[C]}$ and $H_2O^{[F]1}$ reached 100 and 34%, respectively, (Fig. 4a–c; Supplementary Section 15-16, Supplementary Figs. 19–28 and Supplementary

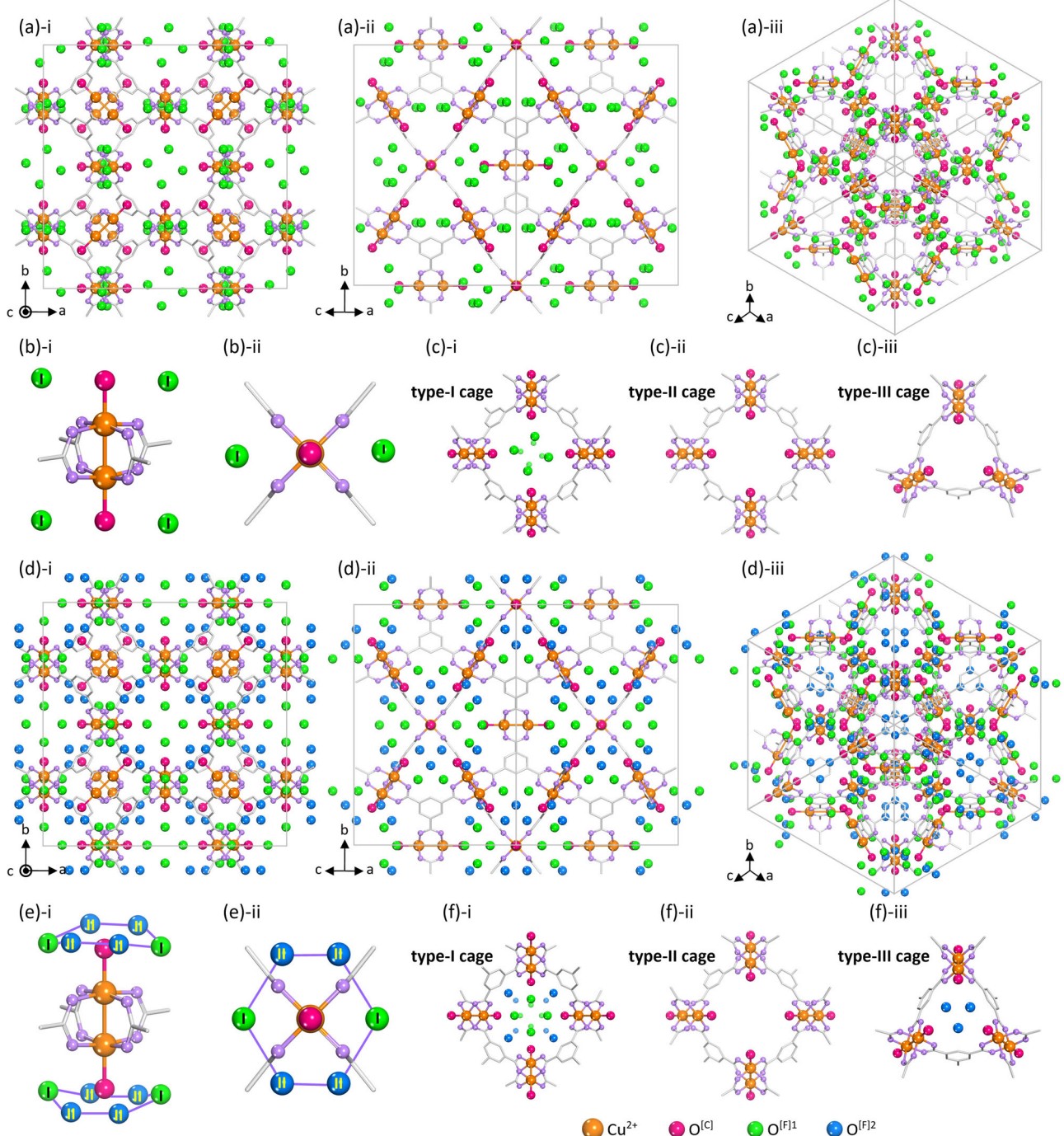

**Fig. 2 SCXRD structure of H₂O-filled HKUST-1 measured at 220 K. a, d** SCXRD structures of **a** $H_2O^{[C]}$-HKUST-1 and **d** $H_2O^{[F]}$-HKUST-1 crystals, viewed along with the (**i**) [001], (**ii**) [101] and (**iii**) [111] directions. **b, e** (**i**) 3-D view and (**ii**) twodimensional (2-D) view of Illustrative spatial configurations of $H_2O^{[C]}$ (red spheres), $H_2O^{[F]1}$ (green spheres) and $H_2O^{[F]2}$ (blue spheres) molecules around the paddlewheel Cu–Cu node in the (**b**) $H_2O^{[C]}$-HKUST-1 and (**e**) $H_2O^{[F]}$-HKUST-1. **c, f** Illustrative spatial arrangements of $H_2O^{[C]}$ (red spheres), $H_2O^{[F]1}$ (green spheres) and $H_2O^{[F]2}$ (blue spheres) molecules around (**i**) type-I, (**ii**) type-II and (**iii**) type-III cages in the **c** $H_2O^{[C]}$-HKUST-1 and **f** $H_2O^{[F]}$-HKUST-1. Red, green and blue spheres indicate oxygen atoms in $H_2O^{[C]}$, $H_2O^{[F]1}$ and $H_2O^{[F]2}$ molecules, respectively. Hydrogen atoms bound to carbon atoms in benzene moieties and water molecules are omitted for the sake of clarity.

Table 5–16; also see original crystal structures in Supplementary Data 1). However, the length was inversely decreased as the exposure time increased further and finally minimised to 2.617 Å when the exposure time was 240 min. When the exposure time was 180 min, type-2 and type-3 pore-filling H₂O molecules (abbreviated as $H_2O^{[F]3}$) began to fill in the type-II cages (Fig. 4d–f). Thus, these in situ crystallographic results agree well

with the in situ Raman results (vide supra). Specifically, the vibrational band was redshifted from 231 to 163 cm⁻¹, simultaneously with the Cu–Cu length elongation from 2.484 Å to 2.632 Å. However, the band was reversely blueshifted from 163 to 176 cm⁻¹, simultaneously with the Cu–Cu shortening from 2.632 to 2.617 Å. Therefore, the H-bond reflects on the Cu–Cu length and its vibrational mode, exhibiting chain connectivity.

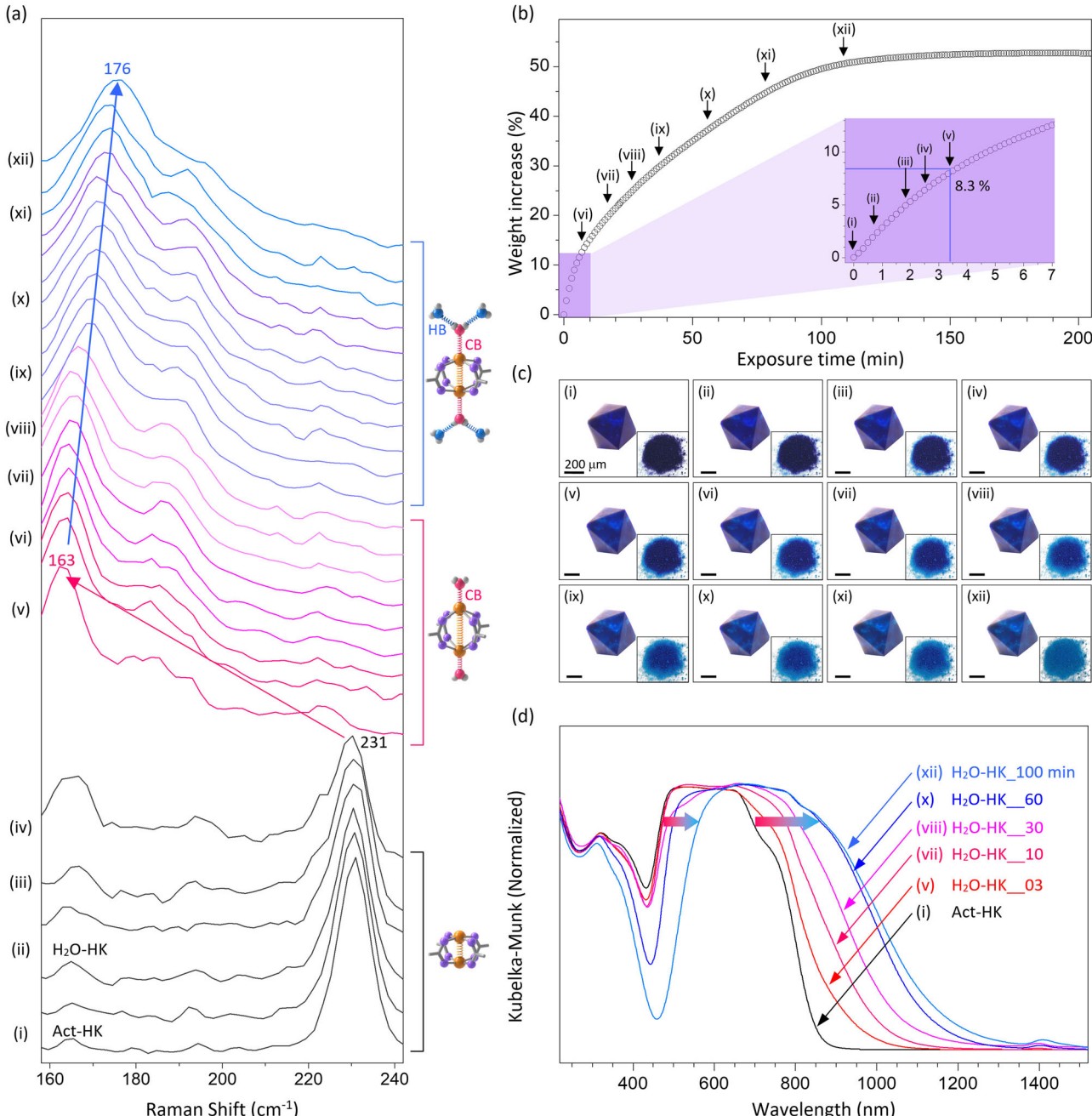

**Fig. 3 In situ Raman study of Cu–Cu vibration. a** Expanded view of in situ confocal micro-Raman spectra of an Act-HKUST-1 crystal recorded at room temperature, exposing the crystal to humid air. **b, c** Successive (**b**) weight and (**c**) colour changes of the Act-HKUST-1 crystal recorded on a microbalance and optical microscope, respectively, exposing the crystal to humid air. Insets in **c** are the photographic images of corresponding powdery crystals. Weight changes, optical microscope images and photograph images of the crystals were examined at the same condition where the confocal micro-Raman spectra were taken. **d** Successive In situ UV-vis absorption spectra of an Act-HKUST-1 crystals measured at room temperature, exposing the crystals to humid air.

Meanwhile, the inclusion of 34 mol% $H_2O^{[F]1}$ can be ascribed to the dynamic coordination bond character of the H-bond, as described above. Based on the single-crystal structure of a fully $H_2O$-filled HKUST-1 (i.e. $H_2O$-HKUST-1($9^{th}$)), we calculated the theoretical numbers of coordinating and pore-filling $H_2O$ molecules in a unit cell. The calculation resulted in the maximum values of 1.00 and $4.66_6$ for the ratios of $[H_2O^{[C]}]/[Cu^{2+}]$ and $[H_2O^{[F]}]/[Cu^{2+}]$, respectively, showing a good agreement with the above water uptake result (Supplementary Section 13 and Supplementary Table 4).

**Distortion of paddlewheel node.** Based on the Cu–Cu length variation after $H_2O$-coordination, we expected that the $O_{BTC}$–Cu–$O_{BTC}$ angle could also be altered, giving rise to a geometrical distortion around the $Cu^{2+}$ centres. Figure 5 shows a pattern that the $O_{BTC}$–Cu–$O_{BTC}$ angle dramatically decreases as the Cu–Cu length increases, with simultaneous changes in lattice parameters (Supplementary Section 17, Supplementary Figs. 29–30 and Supplementary Table 17). We speculate that the increase of Cu–Cu length and subsequent decrease of the bond angle is attributed to the partial electron transfer from the coordinating

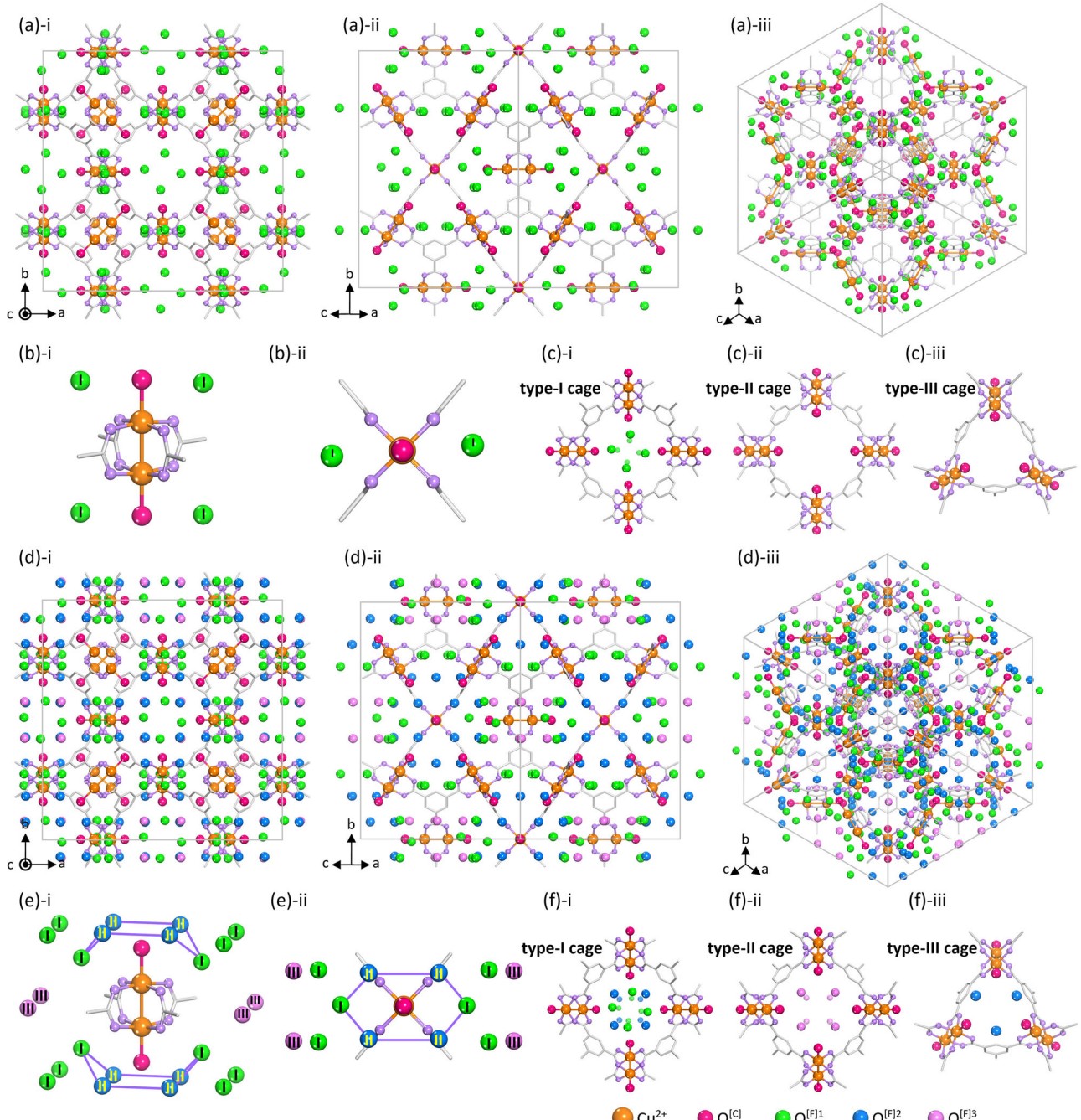

**Fig. 4 In situ SCXRD structure of H₂O-filled HKUST-1 measured at 298 K. a, d** SCXRD structures of the **a** H₂O-HKUST-1(6th) and **d** H₂O-HKUST-1(9th) crystals, viewed along with the (**i**) [001], (**ii**) [101] and (**iii**) [111] direction. SCXRD data were taken by exposing an Act-HKUST-1 single crystal to moist air in the in situ manners (see Supplementary Section 14). **b, e** (**i**) 3-D view and (**ii**) 2-D view of the illustrative spatial configuration of H₂O$^{[C]}$ (red spheres), H₂O$^{[F]1}$ (green spheres), H₂O$^{[F]2}$ (blue spheres) and H₂O$^{[F]3}$ (pink spheres) molecules around the paddlewheel Cu–Cu node in the **b** H₂O-HKUST-1(6th) and **e** H₂O-HKUST-1(9th) crystals. **c, f** Illustrative spatial arrangements of H₂O$^{[C]}$ (red spheres), H₂O$^{[F]1}$ (green spheres), H₂O$^{[F]2}$ (blue spheres) and H₂O$^{[F]3}$ (pink spheres) molecules around (**i**) type-I, (**ii**) type-II and (**iii**) type-III cages in the **c** H₂O-HKUST-1(6th) and **f** H₂O-HKUST-1(9th). Red, green, blue and pink spheres indicate oxygen atoms in H₂O$^{[C]}$, H₂O$^{[F]1}$, H₂O$^{[F]2}$ and H₂O$^{[F]3}$ molecules, respectively. Hydrogen atoms bound to carbon atoms in benzene moieties and water molecules are omitted for the sake of clarity.

H₂O to the antibonding molecular orbitals of Cu²⁺ such as $b_1(x^2-y^2)$ or $a_1(z^2)$[48–50]. A decrease in Cu–O$^{[C]}$ length demonstrates this speculation because the decrease can be attributed to the strong association of coordinating H₂O to Cu²⁺ that can arise from its electron donation (Fig. 5). Nonetheless, we could not observe any pattern in the changes after the substantial formation of H-bond in the samples of H₂O-HKUST-1(6th–9th). Although a more thorough study of the H-bond effect on the O$_{BTC}$–Cu–O$_{BTC}$ angle and Cu–O$^{[C]}$ length is necessary, we speculate that these are ascribed to the only slight alteration of Cu–Cu length after H-bond formation (2.632 and 2.617 Å, respectively).

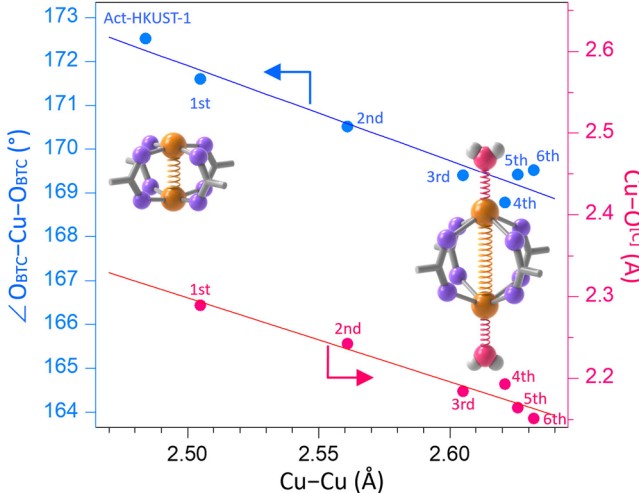

**Fig. 5 Distortion of Cu–Cu paddlewheel centres.** Variation of OBTC–Cu–OBTC angle (blue plot) and Cu–O[C] length (red plot) as a function of Cu–Cu length, whose data were obtained from the in situ synchrotron SCXRD of a moisture-exposed HKUST-1 crystal at 298 K. The linear fit was examined by the method of least squares. The numbers in the figures represent the crystal version from 1st to 6th after exposure to moist air (see Supplementary Section 15 and 16).

## Conclusion

We have discussed the crystalline behaviour of H-bonded water molecules confined in the nanocages of HKUST-1, demonstrating their vibrational chain connectivity and spatial ordering. Based on Raman and SCXRD studies, we have demonstrated that the Cu–Cu length and its vibrational frequency are synchronised together and, by these changes, a substantial distortion arises around the $Cu^{2+}$ centre. The above observations support that the crystalline H-bond can have a dynamic coordination bond character that can dynamically influence the cleavage of previously coordinated water molecules. The above results also support that unbound pore-filling water molecules prefer the sites near water-coordinating $Cu^{2+}$ centers, subsequently forming strong H-bonds (see Supplementary Section 18 and Supplementary Fig. 31). Finally, we envision that these findings—chain connectivities and spatial ordering of H-bonded water molecules, dynamic coordination bond characters of the H-bond, distortion of metal centres and lastly, crystalline behaviour of H-bond—can contribute to not only discovering a more fundamental nature of H-bond in materials but also finding methods to develop efficient porous materials in relevant applications, such as proton conduction, adsorption-based heat pumping and atmospheric water harvesting.

## Methods

**Materials**. All reagents were obtained from commercial sources (Sigma–Aldrich, Alfa Aesar, or Daejung). Copper(II) chloride dihydrate [$CuCl_2 \cdot 2H_2O$, 99%, Sigma–Aldrich], hydrochloric acid (HCl, 37%, Sigma–Aldrich), trimesic acid (1,3,5-bezenetricarboxylic acid, BTC, 95%, Sigma–Aldrich), ethanol (EtOH, 94.5%, Daejung), N,N-dimethylformamide (DMF, 99.5%, Daejung) and distilled deionised water (DDW) were used for the synthesis of HKUST-1 single crystals. Distilled and deionised water (DDW) obtained from a water purification system was used to prepare $H_2O$-containing HKUST-1 crystals. Methanol (MeOH, 99.5%, Daejung) and ethanol (EtOH, 94.5%, Daejung) were used for the preparation of MeOH- and EtOH-containing HKUST-1 crystals, respectively, but these solvents were purified with activated (desolvated) zeolite 4 A in Argon-filled glove box prior to use. Deuterated sulfuric acid ($D_2SO_4$, 96-98 wt% in $D_2O$, 99.5 atomic % in deuterium, Sigma–Aldrich) was used for digesting the HKUST-1 crystals before performing $^1H$-NMR spectroscopic analysis of the samples. All synthesised HKUST-1 crystals were stored in a moisture-free argon-charged glove box before use.

**Synthesis of HKUST-1 single crystals**. We have used large single crystals of HKUST-1 for Raman and SCXRD measurements. The HKUST-1 crystals were synthesised with the same procedure described in our previous work[42,43]. Briefly, $CuCl_2 \cdot 2H_2O$ (1.47 g, 8.6 mmol) was dissolved in 200 mL of DDW using a glass bottle. Then, 2.20 g of DMF was added to the $CuCl_2$ solution. BTC (2.10 g, 10 mmol) was dissolved in 200 mL of EtOH in a separate glass bottle, and 0.25 g of HCl was added into the BTC solution. After the $CuCl_2$ solution was added to the BTC solution bottle, the mixed solution was continuously stirred for 10 min. Then, the bottle was sealed with a polypropylene cap and PTFE tape. The sealed bottle was placed in an oven at 80 °C for 5 d. After the product cooled at room temperature, we collected and washed the pristine HKUST-1 crystals with a mixed solvent of $H_2O$ and EtOH.

**Activation of HKUST-1 crystals**. To purify the as-synthesised HKUST-1 crystals by removing the coordinating and pore-filling $H_2O$ and EtOH molecules, we performed thermal activation with the pristine crystals, using a Schlenk line. For instance, 1.0 g of the pristine HKUST-1 crystals were placed in a glass vacuum tube. Then, the tube was connected to the Schlenk line equipped with a liquid $N_2$ trap-installed vacuum pump. The tube was heated at 150 °C for 24 h under vacuum (~$10^{-3}$ Torr). After cool down to room temperature, the tube was transferred and stored into the moisture-free argon-filled glove box.

**Preparation of $H_2O$-containing HKUST-1 crystals**. We prepared water-coordinating $H_2O^{[C]}$-HKUST-1 crystals by exposing the activated HKUST-1 crystals to moist air. Precisely, the activated HKUST-1 crystals were placed in a disc-shaped quartz cylinder, keeping them exposed to humid air. Then, we monitored the Raman band of Cu–Cu stretching vibration for the sample. When the vibrational band of Act-HKUST-1 at 231 $cm^{-1}$ disappeared utterly, and simultaneously a new band appeared at 165 $cm^{-1}$, we stopped the air exposure by sealing the crystals-containing quartz cylinder with a glass cork. We also prepared water-filled HKUST-1 ($H_2O^{[F]}$-HKUST-1) crystals by sufficiently exposing the Act-HKUST-1 crystals to water vapour with the vial-in-vial method, where a smaller vial containing the Act-HKUST-1 crystals is placed in a capped larger vial containing DDW.

**Preparation of EtOH- and MeOH-containing HKUST-1 crystals**. We prepared $EtOH^{[F]}$-HKUST-1 and $MeOH^{[F]}$-HKUST-1 crystals via soaking Act-HKUST-1 crystals into pure EtOH and MeOH solvents, respectively. Precisely, for the preparation of $EtOH^{[F]}$-HKUST-1, 0.5 g of the Act-HKUST-1 crystals was placed in a glass vial, then 5 mL of EtOH was added into the vial. The vial was sealed with a plastic cap and kept for 2 h. The $MeOH^{[F]}$-HKUST-1 sample was prepared in the same manner except using pure MeOH. EtOH and MeOH were purified with activated zeolite 4 A before use, and all the procedure was performed in moisture-free Ar-charged glove box.

**Sample preparation for the measurement of $^1H$-NMR**. For the $^1H$-NMR measurement of HKUST-1 crystals, a tiny amount of HKUST-1 crystals were placed in an NMR tube, and then 0.6 mL of $D_2SO_4$ was added into the tube. The crystals in the tube were allowed to be completely digested in the sulfuric acid, applying ultrasound in an ultrasonic bath. Finally, the analyte-containing NMR tubes were sealed with plastic caps and acrylic Parafilm® prior to the measurements.

**Water uptake measurements**. To measure the water uptake of HKUST-1 crystals, we spread the Act-HKUST-1 crystals onto a light, shallow plastic container (4 × 4 $cm^2$) placed on top of a microbalance. The initial weight of the crystals was quickly measured, then the increase in the weight of the crystals was continuously monitored at intervals of 10 s, keeping them exposed to the ambient atmosphere (relative humidity of ~30%).

**Sample preparation for Raman and UV-vis absorption spectra measurements**. Before taking Raman and UV-vis absorption spectra, we transferred the HKUST-1 crystals into a moisture-free Ar-charged glove box. Then, the crystals were contained in disc-shaped quartz cells (Starna, Type 37GS Cylindrical Cells with Quartz to Borofloat graded seal). The quartz cells were sealed with a glass cork and grease (Apiezon, H high-temperature Vacuum Greases) before the measurements.

**Ex situ Raman measurements**. A tiny amount of $H_2O^{[F]}$-HKUST-1 crystals were placed in a disc-shaped quartz cell, and the cell was sealed using a glass cork with Apiezon vacuum grease. After taking the initial Raman spectrum with the sample, the cell was connected to a Schlenk vacuum line in order to remove pore-filling $H_2O$ molecules. After applying vacuum to the sample for 5 min, we disconnected the quart cell from the vacuum line and then measured a Raman spectrum. This process was repeated, continuously monitoring Raman spectra. Ex situ Raman spectra of $EtOH^{[F]}$-HKUST-1 and $MeOH^{[F]}$-HKUST-1 crystals were measured, following the same procedure described above.

**In situ Raman measurements**. In a moisture-free Argon-filled glove box, we placed a tiny amount of Act-HKUST-1 crystals in a disc-shaped quartz cell and sealed it with a glass cork and Apiezon vacuum grease. After taking the initial Raman spectrum of Act-HKUST-1, the quartz cell was opened by removing the glass cork in a moist atmosphere. Then, we continuously measured the Raman spectra, keeping the cell open for exposing the HKUST-1 crystal to moist air.

**Sample preparation for SCXRD**. For the ex situ measurements of SCXRD, we prepared Act-HKUST-1, $H_2O^{[C]}$-HKUST-1 and $H_2O^{[F]}$-HKUST-1 samples (*vide supra*), then we transferred the sample to the Ar-charged glove box. Before examining SCXRD measurements, the crystals were kept immersed in oil (Parabar 13012) to prevent their exposure to moisture. In situ SCXRD measurement was performed in a similar manner. An oil-coated Act-HKUST-1 crystal was mounted on the gripper, then the initial diffraction data of Act-HKUST-1 were collected. Continuously removing the coated oil by the nitrogen gas stream in humid air, we collected SCXRD data at 5–30 min intervals.

**Instrumentation**. DDW was obtained from a water purification system (Merck Millipore, MQ Direct 8). $^1$H-NMR spectra were recorded using an AVANCE III HD FT-NMR spectrometer (Bruker, 400 MHz for $^1$H). The $^1$H chemical shifts were referenced to the residual proton resonance of the solvent. Powder X-ray diffraction (PXRD) patterns were obtained using a PANalytical diffractometer (Empyrean) equipped with a monochromatic nickel-filtered Cu Kα beam. Water uptake measurement was performed using a microbalance (Sartorius, Cubis Micro Complete Balance). Solid-state diffuse reflectance UV-vis absorption spectra of samples were recorded using an Agilent Cary 5000 UV-VIS-NIR spectrophotometer. Optical microscope images were taken using an S43T microscope (Bimeince). In situ and ex situ Raman spectra were recorded using a DXRTM2xi Raman Imaging Microscope (Thermo Scientific). Excitation of the samples was performed by focusing a 0.05 mW 532 nm-wavelength laser beam on a crystal with a ×10 magnifying objective lens. Synchrotron single-crystal X-ray diffraction (SCXRD) analyses were performed, using a diffractometer setup with a Pt-coated Si double-crystal monochromator and an ADSC Q210 CCD detector installed in the Pohang Accelerator Laboratory (PAL).

## Data availability

Data supporting the claims and findings of this paper are available within the Supplementary Information or are available upon request from the corresponding author (N.C.J.). Crystallographic data for all structures reported in this paper (shown in Supplementary Data 1) have been deposited with the Cambridge Crystallographic Data Centre (CCDC), CCDC numbers 2091244 for Act-HKUST-1 at 220 K, 2091195 for $H_2O^{[C]}$-HKUST-1 at 220 K, 2091245 for $H_2O^{[F]}$-HKUST-1 at 220 K, 2091247 for Act-HKUST-1 at 298 K, 2091246 for $H_2O$-HKUST-1(1st) at 298 K, 2091254 for $H_2O$-HKUST-1(2nd) at 298 K, 2091260 for $H_2O$-HKUST-1(3rd) at 298 K, 2091258 for $H_2O$-HKUST-1(4th) at 298 K, 2091266 for $H_2O$-HKUST-1(5th) at 298 K, 2091259 for $H_2O$-HKUST-1(6th) at 298 K, 2091261 for $H_2O$-HKUST-1(7th) at 298 K, 2091264 for $H_2O$-HKUST-1(8th) at 298 K and 2091262 for $H_2O$-HKUST-1(9th) at 298 K. Copies of the crystallographic data can be obtained at https://www.ccdc.cam.ac.uk/structures/ free of charge.

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

## Acknowledgements

This work was supported by the Ministry of Science and ICT (MSIT) of Korea under the auspices of the Basic Science Research Program sponsored by the National Research Foundation (NRF-2019R1A2B5B02070032 and NRF-2021R1A4A5030513) and by the DGIST R&D Program (22-HRHR + −0403).

## Author contributions

N.C.J. supervised the project. All authors contributed extensively to the work presented in this paper. J.B. and N.C.J. planned and designed the experimental ideas and methods. J.B. executed the syntheses and characterisation with assistance from S.H.P. D.M. collected and analysed the SCXRD data. The manuscript was written by N.C.J. and J.B. mainly, and all authors have given consent to this publication.

## Competing interests

The authors declare no competing interests.
