## [Peer Review File · Communications Chemistry]

Reviewers' comments:

Reviewer #1 (Remarks to the Author):

The manuscript entitled "Crystalline hydrogen bond of water molecules confined in a metal-organic framework" systematically studied the evolution of water networks in a classical metal-organic framework (MOF) HKUST-1. The authors capitalized their expertise in crystallography and Raman spectroscopy to determine the atom location and bonding in the complicated system. MOFs are emerging porous materials that feature application potentials in gas storage/separation, catalysis, and energy storage. Recently, MOFs have been applied as new sorbents to capture water from the air. However, designing MOFs with distinguished water adsorption capacity requires a deep understanding of the adsorption process, which is dramatically hindered by the structural complexity. Therefore, this work is of fundamental significance, which will draw great interest from various fields, including materials science and sustainable energy. Although some level A alerts were reported for the cif file, their reasons have been addressed by the author. Besides, the manuscript is well written, and minor revisions are recommended before its publication in Communications Chemistry.

Major concern:

1. The authors utilized activated HKUST-1 to study the adsorption of water. But they did not confirm whether the open metal sites of HKUST-1 are occupied by water or not. Herein, FT-IR is highly recommended to study the MOF, which can provide direct evidence of the bonding between water and copper paddle wheel.

Minor Concerns:

1. Very recently, Yaghi and coworkers reported a paper that studied the evolution of water in MOF-303 using crystallographic techniques. (Ref: Science 374, 454–459 (2021)) These two manuscripts are highly related, so I recommend the authors cite and discuss Yaghi's paper.
2. Scale bar is recommended for the optical images in Fig. 3c. If possible, solid-state UV-vis can be added into Figure 3 to confirm the color change of HKUST-1.
3. Powder X-ray diffraction (PXRD) is recommended to monitor the crystallinity of HKUST-1. Besides, the distorted paddlewheel clusters may lead to a shift in lattice parameters, which may be observed in PXRD.

Reviewer #2 (Remarks to the Author):

In this work, Bae et al. have thoroughly demonstrated that not only do changes in the Cu–Cu length in correlate to its vibrational frequency, but consequently, a substantial distortion arises around the Cu²⁺ centre of the paddlewheel unit in HKUST-1. The Raman and SCXRD data also suggest that the hydrogen bond dynamically influences the cleavage of previously coordinated water molecules through chain connectivity and spatial order of the hydrogen bonded water molecules.

This work is novel in that it illustrates how connective chains formed through H-bonding enable spatial ordering of H-bonded water molecules, dynamic coordination bond characters of the H-bond, and distortion of metal centres. Hydrogen bonding of confined water molecules in MOFs has important implications for those working in proton conductivity, adsorption-based heat pumping,

and atmospheric water harvesting.

However, multiple revisions must be incorporated in the introduction:

Lines 53-55: Co^{2+} featured in Dinca's work has lower ionizing electronegativity and thus lower H-bonding capability than Cu^{2+} . Is this a fair comparison for the H-bonding observed in this work?

Line 58: Do unbound water molecules prefer hydrophilic sites in HKUST-1, even though the metal center has Cu^{2+} in the paddlewheel MOF, not saturated Zn^{2+} paddlewheel MOF in Walton's work?

Lines 62-65: The rationale for HKUST-1 selection is not explicitly clear, as written. What criteria are the authors aiming for other than the unique Raman vibrational feature? The intent is unclear until later in the manuscript, so my suggestion is to move Lines 92-119 after Line 65.

Line 68: Start a new paragraph at "This work presents..."

Lines 76-77: The authors claim that "connectivity of H-bonded water molecules underscores that their H-bonds can behave like the strong chemical bond of the crystalline lattice..." Without citing quantitative dissociation energies of H-bonding, which is a highly contested topic, this claim lacks support.

Lines 123-125: These terms need to be defined earlier on MS-3.

Moreover, the following parts of the manuscript need revision:

Line 41: Add "and" after "atmospheric water harvesting,"

Line 47: Change to "the OMS can substantially"...

Line 68-72: The terms used to differentiate between H-bonded H_2O vs. coordinated H_2O in the text misalign with the legend in Figure 1. Please make words consistent between text and legend.

Line 109-110: Change to "the paddlewheel Cu^{2+} can undergo reversible association"

Line 112: Change to "Envisioning a strong association..."

Line 113: Change to "site-selectively occupy the pores"

Line 131: Start "Results & Discussion" section here. Lines 92-119 can go into the Introduction, while

Lines 121-129 seems repetitive since there is already a "Methods" section on MS16-MS17.

Line 137: Remove "certainly".

Line 136-142: These lines are really confusing to follow since there is contradictory logic flow.

Line 141: Remove "of course".

Line 156-157: Remove, as it's repetitive in lines 162-163.

Line 222: Replace "Concretely" with "Specifically".

Line 251: Replace "proving" with "demonstrating".

Line 259: Replace "material beings" with "materials".

Line 260: Briefly reiterate that H-bonding has important implications for those working in proton conductivity, adsorption-based heat pumping, and atmospheric water harvesting, as stated in the introduction.

While the manuscript offers fundamental insight into crystalline H-bonding in HKUST-1 through comprehensive spectroscopic and diffraction studies, it is difficult to follow at times due to the sheer amount of data referenced between the text and supplementary information. The authors are encouraged to evaluate which data types can be omitted for clarity.

Reviewer #3 (Remarks to the Author):

Jeong and colleagues present a thorough study on the structure of water within the pores of Cu-HKUST-1. This is a nice study, and explores two interesting areas of MOF chemistry; 1) the structure

and orientation of guests - specifically water - within the pores of MOFs, and 2) the dynamic behavior of the Cu centers in HKUST-1. In the former, the topic has become increasingly important as MOFs continue to be of interest for cycleable water capture and release. In the latter, the interactions between open metal sites and polar guests is fundamentally interesting and the authors do a nice job of providing vibrational evidence to support their interpretation that the Cu-Cu bonds are affected by axial ligation.

One of the central findings in this paper is captured in Figure 5, showing the distortion of the MOF nodes with axial ligation. The authors could benefit from including some molecular orbital arguments for why this distortion occurs. I also wonder about the implication of the largest pore in HKUST-1 having no open-metal sites pointing into it. How does one rationalize the adsorption isotherm (not presented in the main text), given that water nucleation seems to prefer binding to the open metal site first.

Other than that, this is a nice paper and is a valuable contribution to the field of water storage in MOFs.

Crystalline hydrogen bond of water molecules confined in a metal-organic framework

Jinhee Bae,¹ Sun Ho Park,¹ Dohyun Moon^{2*} and Nak Cheon Jeong^{1*}

¹Department of Emerging Materials Science, DGIST, Daegu 42988, Korea

²Beamline Department, Pohang Accelerator Laboratory, Pohang 37673, Korea

email: dmoon@postech.ac.kr; nc@dgist.ac.kr

ABSTRACT:

Hydrogen bond (H-bond) of water molecules confined in nanopores is of particular interest because it is expected to exhibit chemical features different from bulk water molecules due to its interaction with the wall lining the pores. Herein, we show a crystalline behavior of hydrogen-bonded water molecules residing in the nanocages of a paddlewheel metal-organic framework, providing *in situ* and *ex situ* synchrotron single-crystal X-ray diffraction and Raman spectroscopy studies. The crystalline H-bond is demonstrated by proving the vibrational chain connectivity arising between hydrogen bonding and paddlewheel Cu–Cu bonding in sequentially connected Cu–Cu·····coordinating H₂O·····H-bonded H₂O and proving the spatial ordering of H-bonded water molecules at room temperature, where they are anticipated to be disordered with a high degree of freedom in their molecular motions. Additionally, we show a substantial distortion of the paddlewheel Cu²⁺ centers that arises simultaneously with water coordination. Also, we suggest the dynamic coordination bond character of the H-bond of the confined water molecules, by which an H-bond transitions to a coordination bond at the Cu²⁺ center instantaneously after dissociating a previously coordinated water molecule.

28 INTRODUCTION

[revised manuscript text omitted]

Start R&D section here.

**Raman and crystallography studies at 220 K.** We conducted Raman studies with Act-HKUST-1,
$\text{H}_2\text{O}^{[\text{C}]}$ -HKUST-1 and $\text{H}_2\text{O}^{[\text{F}]}$ -HKUST-1 initially at 220 K. While an Act-HKUST-1 exhibited a Raman
band for Cu–Cu stretching vibration at 231 cm^{-1} , $\text{H}_2\text{O}^{[\text{C}]}$ -HKUST-1 and $\text{H}_2\text{O}^{[\text{F}]}$ -HKUST-1 exhibited the
bands at 166 and 176 cm^{-1} , respectively (Supplementary Section 4). To investigate the correlation
between the Cu–Cu vibrational band energies and their bond lengths upon the HKUST-1 states, we
conducted synchrotron SCXRD experiments with the crystals at 220 K. The SCXRD results show that
the Act-HKUST-1 crystal contains no water molecule **certainly** (Supplementary Section 5). We
expected the $\text{H}_2\text{O}^{[\text{C}]}$ -HKUST-1 to contain coordinating H_2O only because the Cu–Cu Raman band
appeared at the **lowest frequency** (166 cm^{-1}). In contrast to our expectation, however, the $\text{H}_2\text{O}^{[\text{C}]}$ -
HKUST-1 **contained pore-filling H_2O as well with 40 mol% at specific positions in type-I cages**
(denoted as type-1 pore-filling H_2O with the abbreviation of $\text{H}_2\text{O}^{[\text{F}1]}$), **of course**, in addition to

coordinating H₂O (H₂O^[C], see Fig. 2a-c and Supplementary Table 2-3). The appearance of the Raman
band at the lowest frequency despite the inclusion of the H₂O^{[F]1} is ascribed to the *dynamic coordination*
*bond* character of H-bond, by which H₂O^{[F]1} can associate at the Cu²⁺ centre instantaneously after
replacing the previous coordination bond of H₂O^[C] on the molecular time scale⁴⁴⁻⁴⁶. SCXRD result of
H₂O^[F]-HKUST-1, where pore-filling H₂O molecules are supposed to bind at coordinating H₂O via H-
bonds strongly, shows a type of coordinating H₂O and two types of pore-filling H₂O (type-1 and type-2;
denoted with the abbreviation of H₂O^{[F]2}, see Fig. 2d-f). The result describes accurate positions of the

[revised manuscript text omitted]

$[\text{H}_2\text{O}^{[\text{C}]}]/[\text{Cu}^{2+}]$ and $[\text{H}_2\text{O}^{[\text{F}]}]/[\text{Cu}^{2+}]$, respectively, showing a good agreement with the above water
uptake result (Supplementary Table 16).

Fig. 4 | In situ SCXRD structure of H₂O-filled HKUST-1 measured at 298 K. **a** and **d**, SCXRD structures of the **(a)** H₂O-HKUST-1(6th) and **(d)** H₂O-HKUST-1(9th) crystals, viewed along with the **(i)** [001], **(ii)** [101] and **(iii)** [111] direction. SCXRD data were taken by exposing an Act-HKUST-1 single crystal to moist air in the in situ manners (see Supplementary Section 13). **b** and **e**, **(i)** 3-D view and **(ii)** 2-D view of the illustrative spatial configuration of H₂O^[C] (red spheres), H₂O^[F1] (green spheres), H₂O^[F2] (blue spheres) and H₂O^[F3] (pink spheres) molecules around the paddlewheel Cu–Cu node in the **(b)** H₂O-HKUST-1(6th) and **(e)** H₂O-HKUST-1(9th) crystals. **c** and **f**, Illustrative spatial arrangements of H₂O^[C] (red spheres), H₂O^[F1] (green spheres), H₂O^[F2] (blue spheres) and H₂O^[F3] (pink spheres) molecules around **(i)** type-I, **(ii)** type-II and **(iii)** type-III cages in the **(c)** H₂O-HKUST-1(6th) and **(f)** H₂O-HKUST-1(9th). Red, green, blue and pink spheres indicate oxygen atoms in H₂O^[C], H₂O^[F1], H₂O^[F2] and H₂O^[F3] molecules, respectively. Hydrogen atoms bound to carbon atoms in benzene moieties and water molecules are omitted for the sake of clarity.

**Distortion of paddlewheel node.** Based on the Cu–Cu length variation after H₂O-coordination, we

expected that the $O_{BTC}-Cu-O_{BTC}$ angle could also be altered, giving rise to a geometrical distortion

around the Cu^{2+} centres. Fig. 5 shows a pattern that the $\text{O}_{\text{BTC}}\text{-Cu-O}_{\text{BTC}}$ angle dramatically decreases
 as the Cu-Cu length increases. We speculate that the increase of Cu-Cu length and subsequent
 decrease of the bond angle is attributed to the partial electron transfer from the coordinating H_2O to the
 antibonding molecular orbitals of Cu^{2+} such as $b_1(x^2-y^2)$ or $a_1(z^2)$ ⁴⁷⁻⁴⁹. A decrease in $\text{Cu-O}^{[\text{C}]}$ length
 demonstrates this speculation because the decrease can be attributed to the strong association of
 coordinating H_2O to Cu^{2+} that can arise from its electron donation (Fig. 5). Nonetheless, we could not
 observe any pattern in the changes after the substantial formation of H-bond from $\text{H}_2\text{O-HKUST-1}$ (6th).
 Although a more thorough study of the H-bond effect on the $\text{O}_{\text{BTC}}\text{-Cu-O}_{\text{BTC}}$ angle and $\text{Cu-O}^{[\text{C}]}$ length
 is necessary, we speculate that these are ascribed to the only slight alteration of Cu-Cu length after H-
 bond formation (2.632 and 2.617 Å, respectively).

Fig. 5 | Distortion of Cu-Cu paddlewheel centres. Variation of $\text{O}_{\text{BTC}}\text{-Cu-O}_{\text{BTC}}$ angle (blue plot) and (b) $\text{Cu-O}^{[\text{C}]}$ length (red plot) as a function of Cu-Cu length, whose data were obtained from the in situ synchrotron SCXRD of a moisture-exposed HKUST-1 crystal at 298 K. The linear fit was examined by the method of least squares. The numbers in the figures represent the crystal version from 1st to 6th after exposure to moist air (see Supplementary Section 13).

Conclusion

We have discussed the crystalline behaviour of H-bonded water molecules confined in the nanocages of
 HKUST-1, **proving** their vibrational chain connectivity and spatial ordering. Based on Raman and
 softer word - demonstrating

SCXRD studies, we have demonstrated that the Cu–Cu length and its vibrational frequency are
synchronised together and, by these changes, a substantial distortion arises around the Cu²⁺ centre.
The above observations also support that the crystalline H-bond can have a *dynamic coordination bond*
character that can dynamically influence the cleavage of previously coordinated water molecules.
Finally, we envision that these findings—chain connectivities and spatial ordering of H-bonded water
molecules, dynamic coordination bond characters of the H-bond, distortion of metal centres and lastly,
crystalline behaviour of H-bond—can contribute to not only discovering a more fundamental nature of H-
bond in material beings but also finding methods to develop efficient porous materials in relevant
applications.

**Online contents**

Any methods, additional references, Nature Research reporting summaries, source data, extended data, supplementary
information, acknowledgements, peer review information; details of author contributions and competing interests; and
statements of data and code availability are available at <https://doi.org/>.

[revised manuscript text omitted]
 $\text{H}_2\text{O}^{[\text{C}]}$ -HKUST-1 at 220 K, 2091245 for $\text{H}_2\text{O}^{[\text{F}]}$ -HKUST-1 at 220 K, 2091247 for Act-HKUST-1 at 298 K,
462 2091246 for H_2O -HKUST-1(1st) at 298 K, 2091254 for H_2O -HKUST-1(2nd) at 298 K, 2091260 for H_2O -HKUST-1(3rd) at 298
463 K, 2091258 for H_2O -HKUST-1(4th) at 298 K, 2091266 for H_2O -HKUST-1(5th) at 298 K, 2091259 for H_2O -HKUST-1(6th) at
464 298 K, 2091261 for H_2O -HKUST-1(7th) at 298 K, 2091264 for H_2O -HKUST-1(8th) at 298 K and 2091262 for H_2O -HKUST-
465 1(9th) at 298 K. Copies of the crystallographic data can be obtained at <https://www.ccdc.cam.ac.uk/structures/> free of charge.

Acknowledgements

This work was supported by the Ministry of Science and ICT (MSIT) of Korea under the auspices of the Basic Science
Research Program sponsored by the National Research Foundation (NRF-2019R1A2B5B02070032 and NRF-
2021R1A4A5030513) and by the DGIST R&D Program (21-HRHR+-0403).

**Author contribution**
473 N.C.J. supervised the project. All authors contributed extensively to the work presented in this paper. J.B. and N.C.J.
planned and designed the experimental ideas and methods. J.B. executed the syntheses and characterization with assistance
from S.H.P. D.M. collected and analyzed the SCXRD data. The manuscript was written by N.C.J and J.B. mainly, and all
authors have given consent to this publication.

**Competing interests**
The authors declare no competing interests.

**Additional information**
Supplementary information is available for this paper. Correspondence and requests for materials should be addressed to
(Authors information is removed for Double Blind). Reprints and permissions information is available at
www.nature.com/reprints. Publisher's Note Springer Nature remains neutral with regard to jurisdictional claims in
published maps and institutional affiliations.

**Table of Contents (TOC) Artwork**

Spatial ordering of
H-bonded water molecules

Chain-connectivity of
H-bonded water molecules

Graduate School
DGIST

Department of Physics & Chemistry
333, Techno Jungang Daero,
Hyeonpung-Myeon, Dalseong-Gun,
Daegu, 42988, Korea

Nak Cheon Jeong
Associate Professor of Chemistry

nc@dgist.ac.kr
www.nclab.dgist.ac.kr
Office 82-53-785-6513
orcid.org/0000-0003-3320-5750

January 24, 2022

Dear Reviewers,

We appreciate all reviewers' enthusiasm regarding the significance of this work. We have uploaded the Manuscript and Supplementary Information with revisions. The changes are highlighted only in the 'Revised Manuscript for Review Only' and 'Revised Supplementary Information for Review Only'. Below we have detailed our responses to your comments. Briefly, I believe we have responded to all the questions, points of clarification, and requests for the revision.

Sincerely,

Nak Cheon Jeong
Associate Professor of Chemistry

Responses to Reviewers

(See “Revised Manuscript for Review Only” and “Revised Supporting Information for Review Only”)

Reviewer: 1

Comments:

The manuscript entitled “Crystalline hydrogen bond of water molecules confined in a metal-organic framework” systematically studied the evolution of water networks in a classical metal-organic framework (MOF) HKUST-1. The authors capitalized their expertise in crystallography and Raman spectroscopy to determine the atom location and bonding in the complicated system. MOFs are emerging porous materials that feature application potentials in gas storage/separation, catalysis, and energy storage. Recently, MOFs have been applied as new sorbents to capture water from the air. However, designing MOFs with distinguished water adsorption capacity requires a deep understanding of the adsorption process, which is dramatically hindered by the structural complexity. Therefore, this work is of fundamental significance, which will draw great interest from various fields, including materials science and sustainable energy. Although some level A alerts were reported for the cif file, their reasons have been addressed by the author. Besides, the manuscript is well written, and minor revisions are recommended before its publication in Communications Chemistry.

Response:

We immensely appreciate the reviewer for his/her compliments.

Major concern:

(1) The authors utilized activated HKUST-1 to study the adsorption of water. But they did not confirm whether the open metal sites of HKUST-1 are occupied by water or not. Herein, FT-IR is highly recommended to study the MOF, which can provide direct evidence of the bonding between water and copper paddle wheel.

Response:

We appreciate this reviewer for pointing out this interesting one. Water molecule is not present in the ‘activated’ HKUST-1 (Act-HKUST-1); thus, the Act-HKUST-1 does not have any water molecule even coordinating water at open metal sites (OMSs). However, as the Act-HKUST-1 is exposed to humid ambient air, water molecules instantaneously begin to coordinate at the OMSs of the Cu(II) centers because the OMSs are thermodynamically the most stable place for the water molecules in the HKUST-1 with coordination-bond formation energy. The figures 2 and 4 (the results of SCXRD) show the above behavior of water molecules and the Act-HKUST-1. We previously performed Raman studies to observe the bond formation between Cu(II) and water (Cu–O_{H2O}). Please see Supplementary Sections 6, 7, and 11. Typically, the Cu–O bond vibrations appear at approximately 450 – 520 cm⁻¹. However, since the Raman band includes the vibrations of the axial Cu–O_{H2O} bond and the equatorial Cu–O_{COO} framework bond, it is hard to resolve only the Cu–O_{H2O} vibrations from the Raman band. Nevertheless, we were able to observe the Cu–O_{H2O} vibrations from the Raman spectra. Specifically, as the Act-HKUST-1 is exposed to humid air, the band at 503 cm⁻¹ begin to grow up in terms of its intensity, indicating that the amount of coordinating water increases as the exposure time increases. To confirm this water coordination, we have tried to measure IR spectra with the samples of activated and H₂O-ingressed HKUST-1 as this reviewer suggested. Please see the Supplementary Figure 17 in Supplementary Section 12, which is also displayed below. However, we could not observe the Cu–O_{H2O} vibration because of the noisy signal in 450 – 550 cm⁻¹ region with the limitation of reliable energies that our IR equipment affords. Please consider that the measurable energy region of commercial IR equipment is approximately 500 – 4000 cm⁻¹. Instead, we have observed the appearance of broad O–H vibration of water molecules, where the broadness comes from the hydrogen bondings (we speculate these hydrogen bondings are formed between coordinating water and pore-

filling water and between pore-filling waters). The Act-HKUST-1 does not show the broad vibration initially. However, the band begins to appear instantaneously after the exposure to moist air. This pattern we have also observed in Raman spectra. Conclusively, based on the SCXRD, Raman, and IR results, we are convinced that, while Act-HKUST-1 does not include coordinating water, the moisture-exposed HKUST-1 has the coordinating water at the OMSs.

Change:

Addition:

- Manuscript, Line 202: “(also see IR spectra in Supplementary Section S12)”
- Supplementary Information, Page S20, Supplementary Section 12: see below.

Supplementary Section 12. In situ infra-red spectra of an HKUST-1 crystal measured at room temperature.

An insightful reviewer asked us to demonstrate the presence of water in the air-exposed HKUST-1 samples and figure out the coordination of $\text{Cu-O}_{\text{H}_2\text{O}}$. To address these questions, we have tried to observe the Cu-O bond vibrations directly by taking IR spectroscopy, where the vibrations appear at approximately $450 - 520 \text{ cm}^{-1}$. Unfortunately, however, we could not observe the $\text{Cu-O}_{\text{H}_2\text{O}}$ vibration because of the noisy signal in $450 - 550 \text{ cm}^{-1}$ region with the limit of reliable energies capable with commercial IR equipment. Instead, we have observed the appearance of broad O-H vibration of water molecules, where the broadness comes from the hydrogen bondings (we speculate these hydrogen bondings are formed between coordinating water and pore-filling water and between pore-filling waters). Specifically, the Act-HKUST-1 does not show the broad vibration initially, but the band begins to appear instantaneously after the exposure to moist air. This pattern has also been observed in Raman spectra. (See Raman spectra in Supplementary Sections 6, 7, and 11.)

On the other hand, we observed the $\text{Cu-O}_{\text{H}_2\text{O}}$ bond vibrations from the Raman spectra. As the Act-HKUST-1 is exposed to humid air, the band at 503 cm^{-1} begins to grow up in terms of its intensity, indicating that the amount of coordinating water increases as the exposure time increases. (See Supplementary Sections 6, 7, and 11.)

Conclusively, based on the X-ray diffraction results and Raman and IR spectroscopic results, we are convinced that, while Act-HKUST-1 does not include coordinating water, the moisture-exposed HKUST-1 does include the coordinating water at the OMSs.

Supplementary Figure 17. In situ infra-red spectra of activated and H_2O -exposed HKUST-1 crystal samples as indicated.

Minor Concerns:

(1) Very recently, Yaghi and coworkers reported a paper that studied the evolution of water in MOF-303 using crystallographic techniques. (Ref: Science 374, 454–459 (2021)) These two manuscripts are highly related, so I recommend the authors cite and discuss Yaghi's paper.

Response:

We appreciate this reviewer for pointing out this one. We have added the paper as a reference at the introduction part of H-bond dynamics in the main text.

Addition:

– Manuscript, Line 46: a reference (#39) has been added in a sentence of “have rarely been studied^{37–39}.”

(2) Scale bar is recommended for the optical images in Fig. 3c. If possible, solid-state UV-vis can be added into Figure 3 to confirm the color change of HKUST-1.

Response:

Thanks to the reviewer for pointing out this one. We have added a scale bar in every panels of the optical microscope images. Meanwhile, in the original supporting information, we displayed the UV-vis absorption spectra of Act-HKUST-1 and H₂O-exposed HKUST-1 (see Supplementary Section S13). However, as the reviewer suggested, we have newly measured the UV-vis absorption spectra with a fresh Act-HKUST-1 in the in situ manner, simultaneously checking the water uptake contents. Then, we have added the new UV-vis spectra at the panel of Fig. 3d in the main text.

Change:

– Manuscript, Page 11, Figure 3c: scale bars in the optical microscope images have been added. See the figure copied below.

Addition:

– Manuscript, Page 11, Figure 3d: In situ UV-vis absorption spectra of water-exposed HKUST-1 have been added. Also, the figure caption of 3d has been added correspondingly. See the figure copied below.

– Manuscript, Line 220: “Fig. 3d and”

Fig. 3 | In situ Raman study of Cu-Cu vibration. **a**, Expanded view of in situ confocal micro-Raman spectra of an Act-HKUST-1 crystal recorded at room temperature, exposing the crystal to humid air. **b-c**, Successive **(b)** weight and **(c)** colour changes of the Act-HKUST-1 crystal recorded on a microbalance and optical microscope, respectively, exposing the crystal to humid air. Insets in **(c)** are the photographic images of corresponding powdery crystals. Weight changes, optical microscope images and photograph images of the crystals were examined at the same condition where the confocal micro-Raman spectra were taken. **d**, Successive In situ UV-vis absorption spectra of an Act-HKUST-1 crystals measured at room temperature, exposing the crystals to humid air.

(3) Powder X-ray diffraction (PXRD) is recommended to monitor the crystallinity of HKUST-1. Besides, the distorted paddlewheel clusters may lead to a shift in lattice parameters, which may be observed in PXRD.

Response:

We agree with the reviewer for his/her comment. To address the reviewer's question, first, we extracted powder X-ray diffraction (PXRD) patterns from the in situ SCXRD data set, which were previously measured with an activated HKUST-1 crystal, simultaneously exposing it to moist air. From the data, we observed that the diffractions from (220) and (222) planes are most substantially influenced by moisture exposure. To confirm the extracted PXRD, further, we measured PXRD patterns with a fresh powder sample in the in situ manner. As shown below (also see Supplementary Table 16 and Supplementary Figures 29-30), the unit cell length varied upon the moisture exposure. We expected the unit cell length to be linearly correlated or inversely correlated with the Cu–Cu length. However, our observation was different from our expectations. Initially, the unit cell length of the activated HKUST-1 increased as water molecules began to coordinate at the Cu²⁺ center. This point agreed well with our postulation. However, as the adsorbed water became considerable, the changes in the unit cell length did not show any trend. Based on this observation, we speculate that, when pore-filling water molecules begin to increase in the pores, the relation between the unit cell parameter and Cu–Cu length is more complex than what we expected, with substantial factors, such as interactions between different types of pore-filling water molecules and between the pore-filling water molecules and the organic components lining pores. Nonetheless, we guess that the variation (26.243 – 26.399 Å; approximately 0.6% in length) is insufficient to find the relationship between the lattice parameter and paddlewheel length.

Addition:

– Supplementary Information, Page S34–S36: Supplementary Section 16, Supplementary Table 16, Supplementary Figures 29–30 are newly added. Please see below.

Supplementary Section 16. Changes in lattice parameters of HKUST-1.

An insightful reviewer asked us how the lattice parameter of an activated HKUST-1 can be changed during its water sorption. To address the question, we extracted powder X-ray diffraction (PXRD) patterns from the in situ SCXRD data set, which were measured with an activated HKUST-1 crystal, simultaneously exposing it to moist air. Then, we observed that the diffractions from (220) and (222) are most substantially influenced by the degree of moisture exposure. To confirm the extracted PXRD, we also measured PXRD patterns with a powder sample in the in situ manner. As shown in Supplementary Table 16 and Supplementary Figures 29-30 below, the unit cell length varied with respect to the moisture exposure. We expected the unit cell length to be linearly correlated or inversely correlated with the Cu–Cu length. However, our observation was different from our expectations. Initially, the unit cell length of the activated HKUST-1 increased as water molecules began to coordinate at the Cu²⁺ center. This point agreed well with our postulation. However, as the adsorbed water became considerable, the changes in the unit cell length did not show any trend. On the basis of this observation, we speculate that, when pore-filling water molecules begin to increase in the pores, the relation between the unit cell parameter and Cu–Cu length is more complex than what we expected, with many factors, such as interactions between different types of pore-filling water molecules and between the pore-filling water molecules and the organic components lining pores. Nonetheless, we think that the variation (26.243 – 26.399 Å; approximately 0.6% in length) is not enough to find the relationship between the lattice parameter and paddlewheel length.

Supplementary Table 16. Summary of lattice parameters of an Act-HKUST-1 crystal obtained from the in situ SCXRD (see the above Supplementary Tables 14 and 15).

Samples	Exposure time (min)	Unit cell (Å) a = b = c	Δa (Å)	Occupancy of water molecules (mol%)			
				H ₂ O ^{[C]a}	H ₂ O ^{[F]1b}	H ₂ O ^{[F]2b}	H ₂ O ^{[F]3b}
Act-HKUST-1	0	26.249(3)	–	0	0	0	0
H ₂ O-HKUST-1(1 st)	5	26.275(3)	+0.026	10	0	0	0
H ₂ O-HKUST-1(2 nd)	10	26.303(3)	+0.028	76	0	0	0
H ₂ O-HKUST-1(3 rd)	15	26.282(3)	-0.021	96	16	0	0
H ₂ O-HKUST-1(4 th)	20	26.296(3)	+0.014	100	20	0	0
H ₂ O-HKUST-1(5 th)	30	26.243(3)	-0.053	100	28	0	0
H ₂ O-HKUST-1(6 th)	60	26.245(3)	+0.002	100	34	0	0
H ₂ O-HKUST-1(7 th)	120	26.265(3)	+0.020	100	56	0	0
H ₂ O-HKUST-1(8 th)	180	26.393(3)	+0.128	100	56	40	24
H ₂ O-HKUST-1(9 th)	240	26.399(3)	+0.006	100	86	50	18

^aH₂O^[C] is Cu²⁺-coordinating H₂O.

^bH₂O^{[F]1}, H₂O^{[F]2} and H₂O^{[F]3} are type-1, -2 and -3 pore-filling H₂O molecules, respectively.

Supplementary Figure 29. Changes in the lattice parameters of an Act-HKUST-1 single crystal according to the moisture exposure at room temperature.

Projection views of HKUST-1 along [001] and [101] directions

PXRD patterns simulated from SCXRD data

PXRD patterns measured with powder samples

Supplementary Figure 30. (a-b) Projection images of HKUST-1 crystal viewed along (a) [001] and (b) [101] directions. (c) Wide and (d-e) expanded views of PXRD patterns of an Act-HKUST extracted from the in situ SCXRD data set, which were measured by exposing the crystal to moist air continuously. (c) Wide and (d-e) expanded views of in situ PXRD patterns of an activated HKUST-1 powder sample measured, continuously exposing it to moist air.

Reviewer: 2

Comments:

In this work, Bae et al. have thoroughly demonstrated that not only do changes in the Cu–Cu length in correlate to its vibrational frequency, but consequently, a substantial distortion arises around the Cu²⁺ centre of the paddlewheel unit in HKUST-1. The Raman and SCXRD data also suggest that the hydrogen bond dynamically influences the cleavage of previously coordinated water molecules through chain connectivity and spatial order of the hydrogen bonded water molecules.

This work is novel in that it illustrates how connective chains formed through H-bonding enable spatial ordering of H-bonded water molecules, dynamic coordination bond characters of the H-bond, and distortion of metal centres. Hydrogen bonding of confined water molecules in MOFs has important implications for those working in proton conductivity, adsorption-based heat pumping, and atmospheric water harvesting. However, multiple revisions must be incorporated in the introduction:

Response:

We also immensely appreciate the reviewer for his/her compliments.

(1) Lines 53-55: Co²⁺ featured in Dinca's work has lower ionizing electronegativity and thus lower H-bonding capability than Cu²⁺. Is this a fair comparison for the H-bonding observed in this work?

Response:

We agree with the reviewer. While it is difficult to directly compare HKUST-1 with the Dinca's MOF due to their structural difference, we guess that the H-bonding capability of HKUST-1 would be comparable to that of Dinca's MOF because Cu(II) and Co(II) have quite similar electronegativities (ENs). Please note that EN_{Co}=1.91, EN_{Cu}=1.90 in Pauling Scale; EN_{Co}=1.96, EN_{Cu}=1.98 in Sanderson's Scale. For this reason, we commented Dinca's work in the previous version of our manuscript. Please see the original sentences in the main text: "Recently, Paesani, Dincă and coworkers demonstrated the 'templating effect' of H-bond, using a MOF containing cylindrical pores with cobalt OMS³⁷. They observed that water molecules bind to the cobalt OMSs preferentially at low humidity and subsequently form disconnected one-dimensional H-bond chains bridging the cobalt ions at high humidity."

(2) Line 58: Do unbound water molecules prefer hydrophilic sites in HKUST-1, even though the metal center has Cu²⁺ in the paddlewheel MOF, not saturated Zn²⁺ paddlewheel MOF in Walton's work?

Response:

This is a good point. Cu²⁺ centers in HKUST-1 have coordinatively unsaturated 'open-metal sites' where coordinating water molecules can be absent in their activated version. Typically, the surround of transition metal ions is hydrophilic. Also, a metal-coordinating water molecule is usually more hydrophilic than bulk water due to its further polarization by Lewis-acidic transition metal ion. Therefore, the surround of water-coordinated Cu(II) node is estimated to be very hydrophilic, and thus, unbound water molecules residing around Cu(II) node can prefer this hydrophilic site. Our experimental SCXRD results support this estimation, quite close to Walton's results.

Changes:

– Manuscript, Line 87-90:

"However, in situ synchrotron single-crystal X-ray diffraction (SCXRD) results measured at 298 K provide compelling evidence that H-bonded water molecules in nanopores are highly ordered near the coordinated water molecules even at room temperature."

→

“However, in situ synchrotron single-crystal X-ray diffraction (SCXRD) results measured at 298 K provide compelling evidence that unbound H-bonded water molecules in nanopores prefer the sites near the coordinating water molecules with high ordering even at room temperature.”

Addition:

– Manuscript, Line 151-152:

“This result indicates that unbound pore-filling H₂O quite prefers the site near the coordinating H₂O.”

– Manuscript, Line 263-265:

“The above results also support that unbound pore-filling water molecules prefer the sites near water-coordinating Cu²⁺ centers, subsequently forming strong H-bonds (see Supplementary Section 18).”

(3) Lines 62-65: The rationale for HKUST-1 selection is not explicitly clear, as written. What criteria are the authors aiming for other than the unique Raman vibrational feature? The intent is unclear until later in the manuscript, so my suggestion is to move Lines 92-119 after Line 65.

Response:

Thanks a lot for pointing out this one. We agree with the reviewer. However, the main text structure can become complex if we move the hypothesis part to the introduction. Instead, we have corrected the introduction to increase the readability, as the reviewer pointed out. See below.

Change:

– Manuscript, Line 64-71:

“HKUST-1, a paddlewheel MOF also known as Cu₃(BTC)₂, where BTC³⁻ is 1,3,5-benzenetricarboxylate (Fig. 1), have shown a unique Cu–Cu stretching vibrational feature on Raman spectroscopy, depending on the Cu²⁺-coordinating molecular species such as water, methanol (MeOH) and ethanol (EtOH, Supplementary Section 1)⁴⁰⁻⁴⁴.”

→

“HKUST-1, a paddlewheel MOF also known as Cu₃(BTC)₂, where BTC³⁻ is 1,3,5-benzenetricarboxylate (Fig. 1), have shown a unique structural feature with two types of large cages and a type of small cage. The small cages are packed with the fashion of primitive cube, connected by paddlewheel Cu–Cu nodes, and the OMSs at the Cu nodes are faced toward the large cages, where H-bonds between pore-filling and Cu-coordinating water molecules can be formed (See detailed topological description in the “hypothesis” section below). HKUST-1 have also shown a unique Cu–Cu stretching vibrational feature on Raman spectroscopy, depending on the Cu²⁺-coordinating molecular species such as water, methanol (MeOH) and ethanol (EtOH, Supplementary Section 1)⁴⁰⁻⁴⁴.”

(4) Line 68: Start a new paragraph at “This work presents...”

Response:

We appreciate the reviewer for pointing out this one. We have made a new paragraph, as the reviewer suggested.

Change:

– Manuscript, Line 75:

“(Fig. 1e). This work presents”

→

“(Fig. 1e).

¶“This work presents”

(5) Lines 76-77: The authors claim that “connectivity of H-bonded water molecules underscores that their H-bonds can behave like the strong chemical bond of the crystalline lattice...” Without citing quantitative dissociation energies of H-bonding, which is a highly contested topic, this claim lacks support.

Response:

We appreciate the reviewer regarding this point. We have tried to calculate a theoretical value of the H-bond dissociation energy. However, we were unable to obtain the value due to the tremendous number of atoms in this MOF system. Nevertheless, in situ SCXRD results that have shown the crystal-like high ordering of unbound pore-filling water molecules at room temperature and in situ Raman results that have shown vibrational connection behavior of H-bond demonstrate the chemical bond character of H-bond. Instead, we have eliminated “strong” from the sentence because we could not provide the quantitative (even theoretical) value for the H-bond dissociation.

Elimination:

– Manuscript, Line 83: “strong” has been eliminated. See below.

“can behave like the strong chemical bond” → “can behave like the chemical bond”

(6) Lines 123-125: These terms need to be defined earlier on MS-3.

Response:

Thanks for commenting on this point. We understand the reviewer’s comments that the sample abbreviations need to be defined earlier (page 3). But, on page 3, we could not find a suitable place to designate the samples' abbreviation when we considered the whole context. Instead, we have corrected the introduction part on page 3 for better readability with a common sense of broader readership as addressed above (for question 3). See below.

Change:

– Manuscript, Line 64-71:

“HKUST-1, a paddlewheel MOF also known as $\text{Cu}_3(\text{BTC})_2$, where BTC^{3-} is 1,3,5-benzenetricarboxylate (Fig. 1), have shown a unique Cu–Cu stretching vibrational feature on Raman spectroscopy, depending on the Cu^{2+} -coordinating molecular species such as water, methanol (MeOH) and ethanol (EtOH, Supplementary Section 1)⁴⁰⁻⁴⁴.”

→

“HKUST-1, a paddlewheel MOF also known as $\text{Cu}_3(\text{BTC})_2$, where BTC^{3-} is 1,3,5-benzenetricarboxylate (Fig. 1), have shown a unique structural feature with two types of large cages and a type of small cage. The small cages are packed with the fashion of primitive cube, connected by paddlewheel Cu–Cu nodes, and the OMSs at the Cu nodes are faced toward the large cages, where H-bonds between pore-filling and Cu-coordinating water molecules can be formed (See detailed topological description in the “hypothesis” section below). HKUST-1 have also shown a unique Cu–Cu stretching vibrational feature on Raman spectroscopy, depending on the Cu^{2+} -coordinating molecular species such as water, methanol (MeOH) and ethanol (EtOH, Supplementary Section 1)⁴⁰⁻⁴⁴.”

(7) Line 41: Add “and” after “atmospheric water harvesting,”

Response:

Thanks for pointing out this one. But, we have not found any wrong grammar. The phrase after “atmospheric water harvesting” is the one with “which” for describing the formers–proton conduction²⁴, adsorption-based heat pumping²⁵⁻²⁹ and atmospheric water harvesting³⁰⁻³⁶. Please see the original sentence of the previous version of the main text below.

Original Sentence:

“The tunability has recently accelerated the investigation of MOFs for potential use, particularly in proton conduction²⁴, adsorption-based heat pumping²⁵⁻²⁹ and atmospheric water harvesting³⁰⁻³⁶, where H-bond of confined water molecules can play a crucial role.”

(8) Line 47: Change to ”the OMS can substantially”...

Response:

Thanks to the reviewer. We have corrected the sentence by removing “be predicted to”, following the reviewer’s suggestion. See below.

Change:

– Manuscript, Line 49:

“Among these factors, the OMS can be predicted to substantially influence the mobility of pore-filling water molecules”

→

“Among these factors, the OMS can substantially influence the mobility of pore-filling water molecules”

(9) Line 68-72: The terms used to differentiate between H-bonded H₂O vs. coordinated H₂O in the text misalign with the legend in Figure 1. Please make words consistent between text and legend.

Response:

Thanks to the reviewer for pointing out this one. We have slightly corrected the legend placed at the left bottom side in Figure 1. See below

Change:

– Manuscript, Page 5, Figure 1:

“O in type-1 pore-filling H₂O” → “O in type-1 pore-filling (H-bonded) H₂O”

“O in type-2 pore-filling H₂O” → “O in type-2 pore-filling (H-bonded) H₂O”

(10) Line 109-110: Change to “the paddlewheel Cu²⁺ can undergo reversible association”

Response:

Thanks to the reviewer. We have understood the reviewer’s guide for the sentence. We have slightly corrected the sentence from “can be firmly associated together reversely” to “can undergo strong association reversely”. See below.

Change:

– Manuscript, Line 116-117:

“the paddlewheel Cu^{2+} ions can be firmly associated together reversely”

→

“the paddlewheel Cu^{2+} ions can undergo strong association reversely”

(11) Line 112: Change to “Envisioning a strong association...”

Response:

We agree with the reviewer. We have corrected the sentence, following the reviewer’s suggestion. See below.

Change:

– Manuscript, Line 118-119:

“Conceiving a strong association of H-bonded H_2O to the coordinating H_2O ”

→

“Envisioning a strong association of H-bonded H_2O to the coordinating H_2O ”

(12) Line 113: Change to “site-selectively occupy the pores”

Response:

We also agree with the reviewer. We have corrected the sentence, following the reviewer’s suggestion. See below.

Change:

– Manuscript, Line 119-120:

“we further hypothesised that the H-bonded H_2O molecules could be site-selectively sat in the pores”

→

“we further hypothesised that the H-bonded H_2O molecules could site-selectively occupy the pores”

(13) Line 131: Start “Results & Discussion” section here. Lines 92-119 can go into the Introduction, while Lines 121-129 seems repetitive since there is already a “Methods” section on MS16-MS17.

Response:

Thanks for pointing out this one. We agree with the reviewer in part, but we had deeply thought the whole context when we wrote the original manuscript. If we move the early part of ‘Results and Discussion’ to the introduction section, the overall context and assigning the samples abbreviation can be very complex. We are sorry not to accept this recommendation, but we believe the reviewer will understand this difficulty.

(14) Line 137: Remove “certainly”.

(15) Line 136-142: These lines are really confusing to follow since there is contradictory logic flow.

(16) Line 141: Remove “of course”.

Response:

Thanks a lot for pointing out this one. We have thoroughly fixed the sentences for better readability. Please see below.

Change:

– Manuscript, Line 143-152:

“The SCXRD results show that the Act-HKUST-1 crystal contains no water molecule certainly (Supplementary Section 5). We expected the $\text{H}_2\text{O}^{\text{[C]}}$ -HKUST-1 to contain coordinating H_2O only because the Cu–Cu Raman band appeared at the lowest frequency (166 cm^{-1}). In contrast to our expectation, however, the $\text{H}_2\text{O}^{\text{[C]}}$ -HKUST-1 contained pore-filling H_2O as well with 40 mol% at specific positions in type-I cages (denoted as type-1 pore-filling H_2O with the abbreviation of $\text{H}_2\text{O}^{\text{[F]}}$), of course, in addition to coordinating H_2O ($\text{H}_2\text{O}^{\text{[C]}}$, see Fig. 2a-c and Supplementary Table 2-3).”

→

“As expected, the SCXRD results confirm that the Act-HKUST-1 crystal contains no water molecule (Supplementary Section 5). On the other hand, we expected that the $\text{H}_2\text{O}^{\text{[C]}}$ -HKUST-1 would contain coordinating H_2O only because the Cu–Cu Raman band appeared at the lowest frequency (166 cm^{-1} , see Fig. 1 and the hypothesis above). However, the SCXRD result reveals that the $\text{H}_2\text{O}^{\text{[C]}}$ -HKUST-1 contains not only coordinating H_2O (denoted as $\text{H}_2\text{O}^{\text{[C]}}$) but also pore-filling H_2O with 40 mol% at specific positions in type-I cages (denoted as type-1 pore-filling H_2O with the abbreviation of $\text{H}_2\text{O}^{\text{[F]}}$; see Fig. 2a-c and Supplementary Table 2-3). This result indicates that unbound pore-filling H_2O quite prefers the site near the coordinating H_2O ”

(17) Line 156-157: Remove, as it’s repetitive in lines 162-163.

Response:

The sentence on lines 162–163 is not repetitive to the previous lines 156–157. The sentence on the line 162–163 describes the Raman results measured at **170** and **250 K**, while the sentence on the line 156–157 describes the Raman results measured at **298 K**.

Original Sentences:

– Manuscript, Line 165–172:

“As a result, the pattern of spectral changes at 298 K was the same as that observed at 220 K. Whereas the Raman band of $\text{H}_2\text{O}^{\text{[F]}}$ -HKUST-1 appeared at 176 cm^{-1} , the band redshifted approaching 163 cm^{-1} as the filled H_2O s were removed from the pores (Supplementary Section 6). However, the band was blueshifted to 231 cm^{-1} in reverse after all coordinated H_2O s were thoroughly removed from the sample. To ascertain the absence of temperature effect on the vibration, we further performed the Raman analysis at 170 and 250 K with an $\text{H}_2\text{O}^{\text{[F]}}$ -HKUST-1 (Supplementary Section 7). As predicted, the Cu–Cu Raman bands at both 170 and 250 K were the same as those measured at 220 and 298 K.”

(18) Line 222: Replace “Concretely” with “Specifically”.

Response:

We immensely appreciate the reviewer for this suggestion. We have switched “Concretely” to “Specifically”.

Change:

– Manuscript, Line 231: Concretely → Specifically

(19) Line 251: Replace “proving” with “demonstrating”.

Response:

We appreciate the reviewer for this suggestion. We have replaced the word from “proving” to “demonstrating”. See below.

Change:

– Manuscript, Line 259: “proving their vibrational chain connectivity” → “demonstrating their vibrational chain connectivity”

(20) Line 259: Replace “material beings” with “materials”.

Response:

We thank to the reviewer for suggesting this point. We have replaced the word from “material beings” to “materials”. See below.

Change:

– Manuscript, Line 269: “in material beings” → “in materials”

(21) Line 260: Briefly reiterate that H-bonding has important implications for those working in proton conductivity, adsorption-based heat pumping, and atmospheric water harvesting, as stated in the introduction.

Response:

We appreciate the reviewer for this suggestion. We have added some possible applications by reiterating the previous sentence in the introduction section. See below.

Change:

– Manuscript, Line 269-270:

“finding methods to develop efficient porous materials in relevant applications.”

→

“finding methods to develop efficient porous materials in relevant applications, such as proton conduction, adsorption-based heat pumping and atmospheric water harvesting.”

While the manuscript offers fundamental insight into crystalline H-bonding in HKUST-1 through comprehensive spectroscopic and diffraction studies, it is difficult to follow at times due to the sheer amount of data referenced between the text and supplementary information. The authors are encouraged to evaluate which data types can be omitted for clarity.

Response:

We agree with the reviewer on this issue. This paper is somehow difficult to read with lots of data. But, all data we included are evidence for proving our hypothesis. In this regards, we are still hesitating which data set should be eliminated. We will very appreciate this reviewer If the reviewer points out specific data set for the elimination.

Reviewer: 3

Comments:

Jeong and colleagues present a thorough study on the structure of water within the pores of Cu-HKUST-1. This is a nice study, and explores two interesting areas of MOF chemistry; 1) the structure and orientation of guests - specifically water - within the pores of MOFs, and 2) the dynamic behavior of the Cu centers in HKUST-1. In the former, the topic has become increasingly important as MOFs continue to be of interest for cycleable water capture and release. In the latter, the interactions between open metal sites and polar guests is fundamentally interesting and the authors do a nice job of providing vibrational evidence to support their interpretation that the Cu-Cu bonds are affected by axial ligation.

Response:

We also immensely appreciate the reviewer for his/her compliments.

One of the central findings in this paper is captured in Figure 5, showing the distortion of the MOF nodes with axial ligation. The authors could benefit from including some molecular orbital arguments for why this distortion occurs.

Response:

We appreciate the reviewer for this suggestion. We described our speculation on the distortion in terms of molecular orbitals in the previous version of the manuscript. We have attached the original sentence below that describes the molecular orbital.

Original Sentences:

– Manuscript, Line 245–247:

“We speculate that the increase of Cu–Cu length and subsequent decrease of the bond angle is attributed to the partial electron transfer from the coordinating H₂O to the antibonding molecular orbitals of Cu²⁺ such as b₁(x²-y²) or a₁(z²)⁴⁸⁻⁵⁰.”

I also wonder about the implication of the largest pore in HKUST-1 having no open-metal sites pointing into it.

Response:

We thank the reviewer for this point. We have remade Supplementary Figure 2 to show type-II large pores that do not have open-metal sites. Please see Supplementary Figure 2 in Supplementary Information and the figures copied below.

Change:

– Supplementary Information, Page S4, Supplementary Figure 2:

Supplementary Figure 2. (a-c) Illustrations of three types of cages in HKUST-1 and (d) a 2-dimensional (2-D) representation of the topological arrangement of the cages. (e-f) 2-D views of conformational arrangement of Cu nodes in (e) type-I and (f) type-II cages.

How does one rationalize the adsorption isotherm (not presented in the main text), given that water nucleation seems to prefer binding to the open metal site first.

Response:

This is a good point. We completely agree with the reviewer. We have experimented water sorption isotherm with activated HKUST-1. Then, we have found three inflection points in the isotherm that seem to indicate water nucleation in the pores depending on water types, such as coordinating water and type-1, -2, and -3 pore-filling water molecules, respectively. We have made a new Supplementary Section and added the isotherm data with some comments. Please see below and Supplementary Section 18.

Addition:

– Supplementary Information, Page S38, Supplementary Section 18 and Figure 31:

Supplementary Section 18. Water vapor sorption of HKUST-1.

Another insightful reviewer asked us to rationalize the water adsorption isotherm in light of water nucleation around preferred binding sites such as open-metal sites. To address this question, we measured water adsorption isotherm at 25 °C (see Supplementary Figure 31 below). From the isotherm, we found a few inflection points. We ascribe the inflection points to an influence from preferred water sites in pores and correspondingly classified (in this article) water types such as coordinating and type-1, -2, and -3 pore-filling water molecules ($\text{H}_2\text{O}^{[C]}$, $\text{H}_2\text{O}^{[F]1}$, $\text{H}_2\text{O}^{[F]2}$ and $\text{H}_2\text{O}^{[F]3}$, respectively)

Supplementary Figure 31. Water vapor sorption isotherm of an activated HKUST-1 powder sample.

Other than that, this is a nice paper and is a valuable contribution to the field of water storage in MOFs.

Response:

We greatly appreciate the reviewer for his/her compliments.

Additional Corrections

(See “Revised Manuscript for Review Only” and “Revised Supporting Information for Review Only”)

(1) Affiliation

Affiliation for Jinhee Bae, Sun Ho Park, and Nak Cheon Jeong has been changed from “Department of Emerging Materials Science” to “Department of Physics and Chemistry”. Also, a new affiliation for Nak Cheon Jeong (Center for Emerging Materials Science) has been added.

- **Change:** Manuscript, Line 4: “Nak Cheon Jeong^{1*}” → “Nak Cheon Jeong^{1,3*}”
- **Change:** Manuscript, Line 6: “¹Department of Emerging Materials Science” → “¹Department of Physics and Chemistry”
- **Change:** Supplementary Information, Line 16: “Nak Cheon Jeong^{1*}” → “Nak Cheon Jeong^{1,3*}”
- **Change:** Supplementary Information, Line 18: “¹Department of Emerging Materials Science” → “¹Department of Physics and Chemistry”
- **Addition:** Manuscript, Line 9: “³Center for Emerging Materials Science, DGIST, Daegu 42988, Korea”
- **Addition:** Supplementary Information, Line 21: “³Center for Emerging Materials Science, DGIST, Daegu 42988, Korea”

(2) Changes of Reference Number

By adding a reference (#39; see the reviewer 1’s question above on page RL-4), all the reference numbers after 39 have been pushed back.

(3) Changes of Table of Contents in Supplementary Information

We have added Supplementary Section 12, 16, and 18 in the Supplementary Information. Thus, the “Table of Contents” in Supplementary Information has been changed.

- **Addition:** Supplementary Information, Line 54: Supplementary Section 12
- **Addition:** Supplementary Information, Line 68: Supplementary Section 16
- **Addition:** Supplementary Information, Line 70: Supplementary Section 18
- **Change:** Supplementary Information, Line 55: Supplementary Section 12 → Supplementary Section 13
- **Change:** Supplementary Information, Line 56: Supplementary Section 13 → Supplementary Section 14
- **Change:** Supplementary Information, Line 67: Supplementary Section 14 → Supplementary Section 15
- **Change:** Supplementary Information, Line 69: Supplementary Section 15 → Supplementary Section 17

(4) Changes of “Supplementary Section” Number

By addition of Section 12 and 16 in Supplementary Information, the following Section numbers have been changed:

- **Change:** Supplementary Information, Line 884: Supplementary Section 12 → 13
- **Change:** Supplementary Information, Line 922: Supplementary Section 13 → 14

- **Change:** Supplementary Information, Line 1569: Supplementary Section 14 → 15
- **Change:** Supplementary Information, Line 1720: Supplementary Section 15 → 17
- **Change:** Manuscript, Line 220: “Supplementary Section 12” → “Supplementary Section 13”
- **Change:** Manuscript, Line 226-227: “Supplementary Section 13” → “Supplementary Section 14”

(5) Changes of “Supplementary Figure” Number

By addition of Supplementary Figure 17 in Supplementary Information, the following Section numbers have been changed:

- **Change:** Supplementary Information, Line 890: “Fig. 17” → “Figure 18”
- **Change:** Supplementary Information, Line 918: Supplementary Figure 17 → 18
- **Change:** Supplementary Information, Line 966: Supplementary Figure 18 → 19
- **Change:** Supplementary Information, Line 1028: Supplementary Figure 19 → 20
- **Change:** Supplementary Information, Line 1090: Supplementary Figure 20 → 21
- **Change:** Supplementary Information, Line 1152: Supplementary Figure 21 → 22
- **Change:** Supplementary Information, Line 1214: Supplementary Figure 22 → 23
- **Change:** Supplementary Information, Line 1276: Supplementary Figure 23 → 24
- **Change:** Supplementary Information, Line 1338: Supplementary Figure 24 → 25
- **Change:** Supplementary Information, Line 1401: Supplementary Figure 25 → 26
- **Change:** Supplementary Information, Line 1463: Supplementary Figure 26 → 27
- **Change:** Supplementary Information, Line 1525: Supplementary Figure 27 → 28

(6) Changes of “Supplementary Table” Number

By addition of Supplementary Table 16 in Supplementary Information, the following Section numbers have been changed:

- **Change:** Supplementary Information, Line 1729: Supplementary Table 16 → 17
- **Change:** Manuscript, Line 240: “Supplementary Table 16” → “Supplementary Table 17”

(7) Additional Changes

- **Change:** Manuscript, Line 23: “and proving” → “and by proving”
- **Change:** Manuscript, Line 25–26: “arises simultaneously with water coordination” → “arises with water coordination simultaneously”
- **Change:** Manuscript, Line 77: “which must generally appear in a crystalline lattice” → “which must appear in a crystalline lattice generally”
- **Change:** Manuscript, Line 79: “to” → “at”
- **Change:** Manuscript, Line 158: “The” → “This”
- **Change:** Manuscript, Line 250: “from H₂O-HKUST-1(6th)” → “in the samples of H₂O-HKUST-1(6th–9th)”
- **Change:** Supplementary Information, Line 888–889: “see Figure 3c in the text and Supplementary Fig. 16” → “see Fig. 3c in the text and Supplementary Figure 16”
- **Change:** Manuscript, Line 489–502: Additional information in the manuscript has been changed. See below

Additional information

Supplementary information is available for this paper. Correspondence and requests for materials should be addressed to (Authors information is removed for Double Blind). Reprints and permissions information is available at www.nature.com/reprints. Publisher's Note Springer Nature remains neutral with regard to jurisdictional claims in published maps and institutional affiliations.

→

Additional information

Supplementary information The online version contains supplementary material available at <https://doi.org/> is available for this paper.

Correspondence and requests for materials should be addressed to D.M. and N.C.J.

Peer review information Communications Chemistry thanks the anonymous reviewers for their contribution to the peer review of this work.

Reprints and permissions information is available at <http://www.nature.com/reprints>.

Publisher's Note Springer Nature remains neutral with regard to jurisdictional claims in published maps and institutional affiliations.

(8) Additional Eliminations

- **Elimination:** Manuscript, Line 262: “also” has been eliminated.
- **Elimination:** Supplementary Information, Line 113: “of” has been eliminated.

REVIEWERS' COMMENTS:

Reviewer #1 (Remarks to the Author):

In the first round of revision, the authors provided solid experimental evidence, including IR, PXRD, and UV-vis, and made significant revisions to the manuscript. Herein, I strongly recommend the acceptance of the manuscript.

Besides, the authors also mentioned the difficulty in analyzing the fingerprint region of FT-IR. Herein, I recommend one recent publication (<https://doi.org/10.1021/jacs.1c01408>), in which the difference between two IR spectra can be calculated through subtraction. In this way, the IR spectra can provide essential and precise information about the chemical bond in the MOF.

Reviewer #2 (Remarks to the Author):

Thank you for addressing almost all of the revisions. I only have one more to request at this point before publication:

1. Change manuscript, Line 116-117:

“the paddlewheel Cu²⁺ ions can undergo strong association reversely”

to

“the paddlewheel Cu²⁺ ions can undergo strong association reversibly”

Reviewer #3 (Remarks to the Author):

The authors have done a wonderful job of improving the manuscript and it should be accepted.

Graduate School
DGIST

Department of Physics & Chemistry
333, Techno Jungang Daero,
Hyeonpung-Myeon, Dalseong-Gun,
Daegu, 42988, Korea

Nak Cheon Jeong
Associate Professor of Chemistry

nc@dgist.ac.kr
www.nclab.dgist.ac.kr
Office 82-53-785-6513
orcid.org/0000-0003-3320-5750

February 21, 2022

Dear Reviewers,

We appreciate all reviewers' enthusiasm regarding the significance of this work. We have uploaded the Manuscript and Supplementary Information with revisions. The changes are highlighted only in the 'Revised Manuscript for Review Only' and 'Revised Supplementary Information for Review Only'. Below we have detailed our responses to your comments. Briefly, I believe we have responded to all the questions, points of clarification, and requests for the revision.

Sincerely,

Nak Cheon Jeong
Associate Professor of Chemistry

Responses to Reviewers

(See “Revised Manuscript for Review Only” and “Revised Supporting Information for Review Only”)

Reviewer: 1

Comments:

In the first round of revision, the authors provided solid experimental evidence, including IR, PXRD, and UV-vis, and made significant revisions to the manuscript. Herein, I strongly recommend the acceptance of the manuscript.

Response:

We immensely appreciate the reviewer for his/her compliments.

Besides, the authors also mentioned the difficulty in analyzing the fingerprint region of FT-IR. Herein, I recommend one recent publication (<https://doi.org/10.1021/jacs.1c01408>), in which the difference between two IR spectra can be calculated through subtraction. In this way, the IR spectra can provide essential and precise information about the chemical bond in the MOF.

Response:

We thank the reviewer for pointing out this one. As the reviewer suggested, we read the paper (JACS) carefully and then treated our IR data by subtracting the Act-HKUST-1 spectrum from each H₂O-HKUST-1 spectrum (10, 30, and 60 min), respectively. From the difference spectra, we found two vibrational bands appeared at approximately 550 and 595 cm⁻¹. Therefore, we have added the difference spectra and correspondingly some comments in Supplementary Section 12 of the Supplementary Information, and also, we have cited the JACS paper as a reference in both Supplementary Information and the main manuscript.

Additions and Changes:

- 1) We have cited the recent JACS paper, which the reviewer suggested, in the main manuscript: Ref 52.
 1. Manuscript, Line 199: “(also see IR spectra in Supplementary Section S12)⁵².”
 2. Manuscript, Line 461: “52. Tan, K. *et al.* Defect Termination in the UiO-66 Family of Metal–Organic Frameworks: The Role of Water and Modulator. *J. Am. Chem. Soc.* **143**, 6328-6332 (2021).”
- 2) We have cited the recent JACS paper, which the reviewer suggested, in the Supplementary Information: Ref S9.
 1. Supplementary Information, Page S20, Line 844: “in the 500 – 600 cm⁻¹ region^{S9}.”
 2. Supplementary Information, Page S39, Line 1801: “Tan, K. *et al.*, Defect Termination in the UiO-66 Family of Metal–Organic Frameworks: The Role of Water and Modulator. *J. Am. Chem. Soc.* **143**, 6328-6882 (2021).”
- 3) Supplementary Information, Page S20, Supplementary Section 12: We have corrected two sentences.

“Instead, we have observed the appearance of broad O–H vibration of water molecules, where the broadness comes from the hydrogen bondings (we speculate these hydrogen bondings are formed between coordinating water and pore-filling water and between pore-filling waters).”

→

“Instead, we subtracted the Act-HKUST-1 spectrum from each H₂O-HKUST-1 (10, 30, and 60 min) spectrum, respectively, to find a difference in the 500 – 600 cm⁻¹ region^{S9}. Then, we found two bands at approximately 550 and 595 cm⁻¹. Thus, we ascribe the two bands to the vibration of Cu–O_{H2O}. Also, we have observed the appearance of broad O–H vibration of water molecules, where the broadness comes from the hydrogen bondings (we speculate these hydrogen bondings are formed between coordinating water and pore-filling water and between pore-filling waters).”

4) Supplementary Information, Page S20, Supplementary Section 12: Supplementary Figure 17(b) has been added, as seen below.

Reviewer: 2

Comments:

Thank you for addressing almost all of the revisions.

Response:

We immensely appreciate the reviewer for his/her compliments.

I only have one more to request at this point before publication:

Change manuscript, Line 116-117: “the paddlewheel Cu²⁺ ions can undergo strong association reversely” to “the paddlewheel Cu²⁺ ions can undergo strong association reversibly”

Response:

This point is also good. We have corrected “reversely” to “reversibly”.

Change:

– Manuscript, Line 114: “reversely” → “reversibly”

Reviewer: 3

Comments:

The authors have done a wonderful job of improving the manuscript and it should be accepted.

Response:

We immensely appreciate this reviewer as well for his/her compliments.